# Activation of Nrf2/HO-1 signaling pathway exacerbates cholestatic liver injury
Yi Wang [1,2,6], Xiaolong Fu [1,2,6], Li Zeng[1,2], Yan Hu[1,2], Rongyang Gao[1,2], Siting Xian[1,2], Songjie Liao[1,2], Jianxiang Huang[1,2], Yonggang Yang[1,2], Jilong Liu[3], Hai Jin[4], James Klaunig[5], Yuanfu Lu [1,2] ✉ & Shaoyu Zhou [1,2] ✉

Nuclear factor erythroid 2-related factor-2 (Nrf2) antioxidant signaling is involved in liver protection, but this generalization overlooks conflicting studies indicating that Nrf2 effects are not necessarily hepatoprotective. The role of Nrf2/heme oxygenase-1 (HO-1) in cholestatic liver injury (CLI) remains poorly defined. Here, we report that Nrf2/HO-1 activation exacerbates liver injury rather than exerting a protective effect in CLI. Inhibiting HO-1 or ameliorating bilirubin transport alleviates liver injury in CLI models. Nrf2 knockout confers hepatoprotection in CLI mice, whereas in non-CLI mice, Nrf2 knockout aggravates liver damage. In the CLI setting, oxidative stress activates Nrf2/HO-1, leads to bilirubin accumulation, and impairs mitochondrial function. High levels of bilirubin reciprocally upregulate the activation of Nrf2 and HO-1, while antioxidant and mitochondrial-targeted SOD2 overexpression attenuate bilirubin toxicity. The expression of Nrf2 and HO-1 is elevated in serum of patients with CLI. These results reveal an unrecognized function of Nrf2 signaling in exacerbating liver injury in cholestatic disease.

The early stage of cholestasis is usually asymptomatic. After the progression of cholestasis, bile components such as cholic acid and bilirubin accumulate excessively in the liver and systemic circulation, resulting in damage to liver cells and further progression to liver fibrosis and even cirrhosis[1,2]. During cholestasis, the abnormal increase of bilirubin in the body leads to hyperbilirubinemia, which can attack various tissues and organs and cause further deterioration of liver diseases[3]. However, although elevated bilirubin is considered as an indicator of the severity of cholestasis, the hepatoxicity and mechanism of high bilirubin in the liver remain poorly elucidated. In fact, bilirubin has complex biological functions. It is generally believed that bilirubin acts as an antioxidant through biosynthetic cycle from biliverdin by biliverdin reductase[4]. However, research results on the supposed protective effect of bilirubin on oxygen-free radical diseases in the past decades have been conflicting, and the experimental paradigm on the role of bilirubin-biliverdin redox cycling in cell antioxidant protection has also been challenged[5,6]. On the other hand, the toxicity of bilirubin has been well documented, although the underlying mechanisms remain to be further defined[7,8].

Nuclear factor erythroid 2-related factor 2 (Nrf2) is a transcription factor that plays a critical role in cellular defense against toxic and oxidative insults through regulating a battery of antioxidant proteins. Activation of Nrf2 by pharmacological and genetic approaches has been widely involved in the protection of various liver disorders or hepatoxicity[9,10]. Although the activation of Nrf2 has beneficial effects, research in transgenic or Nrf2 knockout ($Nrf2^{-/-}$) mice have led to conflicting reports on the role of Nrf2 in liver protection, indicating that Nrf2 deficiency does not increase liver damage and can even protect the liver from damage[11,12]. In addition, chemical or drug-induced hepatotoxicity is frequently associated with the upregulation of Nrf2, which is considered to be caused by compensation or other undetermined mechanisms against hepatotoxicity. For example, administration of mice with acetaminophen causes liver injury, while it upregulates the gene and protein expressions of Nrf2[13]. Activation of Nrf2 has also been reported in carbon tetrachloride ($CCl_4$)-induced acute liver injury[14]. Increased expression of Nrf2 has also been found in the liver of mice with cholestatic liver injury modeled with oleanolic acid (OA)[15]. However, the exact roles of Nrf2 in these situations have not yet been fully defined.

Heme oxygenase-1 (HO-1) is an antioxidant factor regulated by Nrf2, which plays a protective role against oxidative stress. HO-1 degrades heme to carbon monoxide, free iron, and biliverdin, and biliverdin is further

---

[1]Key Laboratory of Basic Pharmacology of Ministry of Education and Joint International Research Laboratory of Ethnomedicine of Ministry of Education, Zunyi Medical University, Zunyi, China. [2]School of Pharmacy, Zunyi Medical University, Zunyi, China. [3]Department of Gastroenterology, Digestive Disease Hospital, Affiliated Hospital of Zunyi Medical University, Zunyi, China. [4]Institute of Digestive Diseases of Affiliated Hospital, Affiliated Hospital of Zunyi Medical University, Zunyi, China. [5]Department of Environmental and Occupational Health, School of Public Health, Indiana University, Bloomington, IN, USA. [6]These authors contributed equally: Yi Wang, Xiaolong Fu. ✉e-mail: luyuanfu2000@163.com; szhou@zmu.edu.cn

degraded to bilirubin that has anti-oxidative properties. Research has found that bilirubin as low as 10 nM can protect cells against damage up to 10,000 times the concentration of hydrogen peroxide[16]. However, when elevated abnormally, bilirubin may exert toxic effects in various tissues and organs including liver. Intricately, oxidative stress may also play a role in bilirubin-mediated cytotoxicity. For example, exposure to a toxic concentration of unconjugated bilirubin (140 nM) causes increased intracellular reactive oxygen species (ROS) levels along with induction of Nrf2 in neuroblastoma cells, whereas pretreatment with antioxidant N-acetylcysteine significantly reduces oxidative stress and adaptive antioxidant response[17]. In the past few decades, although the elevation of bilirubin in the blood and liver has been widely documented in animal and human cholestatic liver disease, the exact role and potential mechanism of bilirubin elevation in cholestatic liver injury remain unclear.

In the present study, we revealed that activation of Nrf2 resulted in increased expression of HO-1, which was detrimental, while inhibition of HO-1 was protective, in cholestatic liver injury modeled with OA or α-naphthylisothiocyanate (ANIT). $Nrf2^{-/-}$ mice displayed significantly less liver damage in the CLI model, contrary to the results in CCl$_4$-induced non-cholestatic liver injury model in which Nrf2 knockout and inhibition of HO-1 both exacerbated liver injury. Increased expression of $Nrf2$ and $HO-1$ in serum was correlated clinically with the bilirubin content and the severity of patients with cholestatic liver injury. Activation of Nrf2/HO-1 led to excessive accumulation of bilirubin, which in turn damaged mitochondrial function, forming a mutual regulation between these two events, in which mitochondrial ROS played a key role. Both antioxidant and mitochondrial-targeted superoxide dismutase 2 (SOD2) overexpression ameliorated bilirubin-induced hepatotoxicity. These findings demonstrate that the activation of Nrf2 exacerbates liver damage in CLI, and the intervention of Nrf2/HO-1 signaling possesses a potential translational value in the clinical treatment of cholestatic liver disease.

## Results

### Increased HO-1 expression plays a role in pathological changes in CLI

To investigate the role of HO-1 in CLI, we first established a mice model of CLI induced by intragastric administration of OA[18], and found that OA induced extensive hepatocyte necrosis, nucleolysis, and neutrophil infiltration, along with increased alanine aminotransferase (ALT) and aspartate aminotransferase (AST) levels in the serum (Fig. 1a–c). OA treatment also caused significant elevations in alkaline phosphatase (ALP), total bile acid (TBA), as well as total bilirubin (TBIL) and direct bilirubin (DBIL), indicating that OA could induce CLI (Fig. 1d–g). We then analyzed the expression of HO-1 and found that OA significantly increased the protein and mRNA levels of HO-1 (Fig. 1i–k). In primary cultured hepatocytes and $AML12$ cells, OA was also found to cause cellular injury, accompanied by increased levels of HO-1 protein and mRNA expression (Supplementary Fig. 1 and Supplementary Fig. 2). Thus, we proposed that increased HO-1 played a detrimental role in the pathology of CLI modeled with OA. To test this hypothesis, proto-porphyrin IX zinc (ZnPP), a HO-1 inhibitor, was employed before intragastric administration of OA. The administration of animals with the HO-1 inhibitor ZnPP resulted in: (1) significant alleviation of liver tissue injury (Fig. 1a), along with a decrease in serum levels of ALT, AST, and ALP (Fig. 1b–d); and (2) a dramatic decrease in TBIL and DBIL contents (Fig. 1f, g), while no change in TBA level (Fig. 1e). The above results indicate that HO-1 inhibitors can improve OA induced liver injury, which is consistent with previous reports[19,20]. Notably, HO-1 is the rate limiting enzyme for bilirubin synthesis, and the inhibition of HO-1 resulted in a significantly decrease in bilirubin synthesis. Yet, protective effects of HO-1 have been widely described in various disorders including liver injuries such as alcohol-dependent liver damage and other drugs or chemicals induced liver injuries, and most studies on the protective mechanism of HO-1 have been focused on antioxidant stress[21–23]. Accordingly, we hypothesized that induction of HO-1 is

hepatoprotective in the non-CLI setting but hepatotoxic in the CLI model. To test this hypothesis, we further examined the effect of HO-1 on liver injury in a different mouse CLI model induced by ANIT[24]. Treatment with ANIT caused significant liver cholestasis, while HO-1 intervention reduced liver injury and bilirubin accumulation. Multiple focal hepatocyte necrosis and neutrophil infiltration were observed in the ANIT group, along with elevated serum ALT and AST levels, as well as elevated ALP, TBIL, and DBIL levels and liver unconjugated bilirubin content (Fig. 1b–h). In addition, ANIT significantly increased the protein level of HO-1 (Fig. 1l), and the hepatotoxicity induced by ANIT was significantly alleviated after administration of ZnPP, manifested by an improved morphology showing well-arranged hepatic lobules and hepatic cords with only a small amount of inflammation and hepatocyte necrosis (Fig. 1a), along with lower levels in ALT and AST and a dramatic decrease in ALP, TBIL, DBIL and liver unconjugated bilirubin contents (Fig. 1b–d, f–h). All these results were similar to those observed in the CLI model induced by OA. With one exception, the TBA level decreased after inhibition of HO-1 (Fig. 1e), which was inconsistent with the results of the OA model. This may be due to the different degrees of liver damage or the regulatory effect between bilirubin and bile acids[25–27]. Then, we employed a CCl$_4$-induced liver injury model to further characterize the effect of HO-1 on liver injury. CCl$_4$ caused degeneration and necrosis of cells in the portal area or around the central vein (Supplementary Fig. 3a), along with increased serum ALT and AST levels (Supplementary Fig. 3b, c), which was consistent with previous reports[28,29]. CCl$_4$ treatment also resulted in an increase in protein and mRNA levels of HO-1 (Supplementary Fig. 3h, i), but there were no significant changes in the levels of ALP, TBA, TBIL, and DBIL (Supplementary Fig. 3d–g), indicating that CCl$_4$ induced non-CLI. However, CCl$_4$-induced hepatotoxicity was further exacerbated after administration of ZnPP, which was manifested as increased hepatocellular necrotic areas and extensive ballooning degeneration (Supplementary Fig. 3a), with significant elevations in ALT and AST levels (Supplementary Fig. 3b, c). Thus, HO-1 exhibited a completely opposite effect on non-CLI, that is, it had a protective effect on liver cell damage in the non-CLI model.

### Role of Nrf2-dependent regulation of HO-1 in CLI

To investigate the role of Nrf2/HO-1 mediated signal pathways in the pathogenesis of CLI, $Nrf2$ was silenced using short interfering RNA (siRNA) in $AML12$ liver cells (Fig. 2a), and it was found that OA significantly increased the protein expression of Nrf2 and HO-1 in normal cells, while in $Nrf2$-silencing cells, the effect of OA on the expression of Nrf2 and HO-1 proteins was obviously abrogated (Fig. 2b). These results showed that Nrf2/HO-1 signaling pathway was associated with OA induced hepatotoxicity. We further used a $Nrf2^{-/-}$ mice to verify the effect of Nrf2/HO-1 in the CLI. The $Nrf2^{-/-}$ mice were genotyped, and the results showed that the Nrf2 knockout allele sequence had two missing base pairs (GA) between 340 and 350, resulting in code transfer and early termination (Supplementary Fig. 4). As illustrated in the experimental scheme (Fig. 2c), both WT and $Nrf2^{-/-}$ mice were treated with OA and fed with normal diet. While the WT mice exhibited impairments in hepatic metabolism, including mice gallbladder weight, and liver index, $Nrf2^{-/-}$ mice showed significantly less liver impairment (Fig. 2e, f). Next, we evaluated the mice model of OA-induced CLI, and found that compared with WT mice, OA induced liver injury was significantly reduced in $Nrf2^{-/-}$ mice (Fig. 2d). Meanwhile, OA induced significant elevation of liver serum biochemical indicators (ALT, AST, ALP, TBA, TBIL and DBIL) in WT mice. However, in $Nrf2^{-/-}$ mice, the elevation of these liver biochemical indicators induced by OA was significantly inhibited (Fig. 2g–l). In addition, Nrf2 knockout significantly blocked the regulatory effect of OA on HO-1, and was more effective than inhibiting HO-1 in reducing bilirubin accumulation and improving liver damage. On the contrary, when exposed to CCl$_4$ (10 μL/kg), $Nrf2^{-/-}$ mice exhibited significantly more extensive ballooning degeneration compared to WT mice

**Fig. 1 | Increased HO-1 expression is involved in pathological changes in CLI. a** Representative H&E staining of liver tissues in mice (6- to 8-week-old) after administration of ZnPP, followed by 295.9 mg/kg OA or 60 mg/kg ANIT ($n = 5$). Neutrophil infiltration (green arrows) and hepatocyte necrosis (yellow arrows or dashed circles). Scale bar, 100 μm or 50μm. **b–g** Serum levels of ALT, AST, ALP, TBA, TBIL, and DBIL ($n = 5$). **h** Liver UCB concentration ($n = 5$). **i, j** Expression of HO-1 protein and mRNA after oral administration of OA ($n = 6$). **k** Immunohistochemical staining of liver HO-1. Scale bar, 50 μm. Kupffer cells (green triangles) and hepatocyte (yellow triangles). **l** Expression of HO-1 protein after intragastric administration of ANIT ($n = 5$). Student's t test or nonparametric tests (**b–h**), and one-way ANOVA (**h–j**) with Tukey's post hoc test were used for data analysis. All data are shown as Mean ± SEM. *$p < 0.05$, **$p < 0.01$, significant difference compared to the control group; #$p < 0.05$, ##$p < 0.01$, significant difference between the OA or ANIT alone group and the ZnPP-combined group. ALP alkaline phosphatase, ALT alanine aminotransferase, ANIT α-naphthylisothiocyanate, AST aspartate aminotransferase, DBIL direct bilirubin, H&E hematoxylin and eosin, HO-1 heme oxygenase-1, OA oleanolic acid, TBA total bile acid, TBIL total bilirubin, ZnPP protoporphyrin IX zinc.

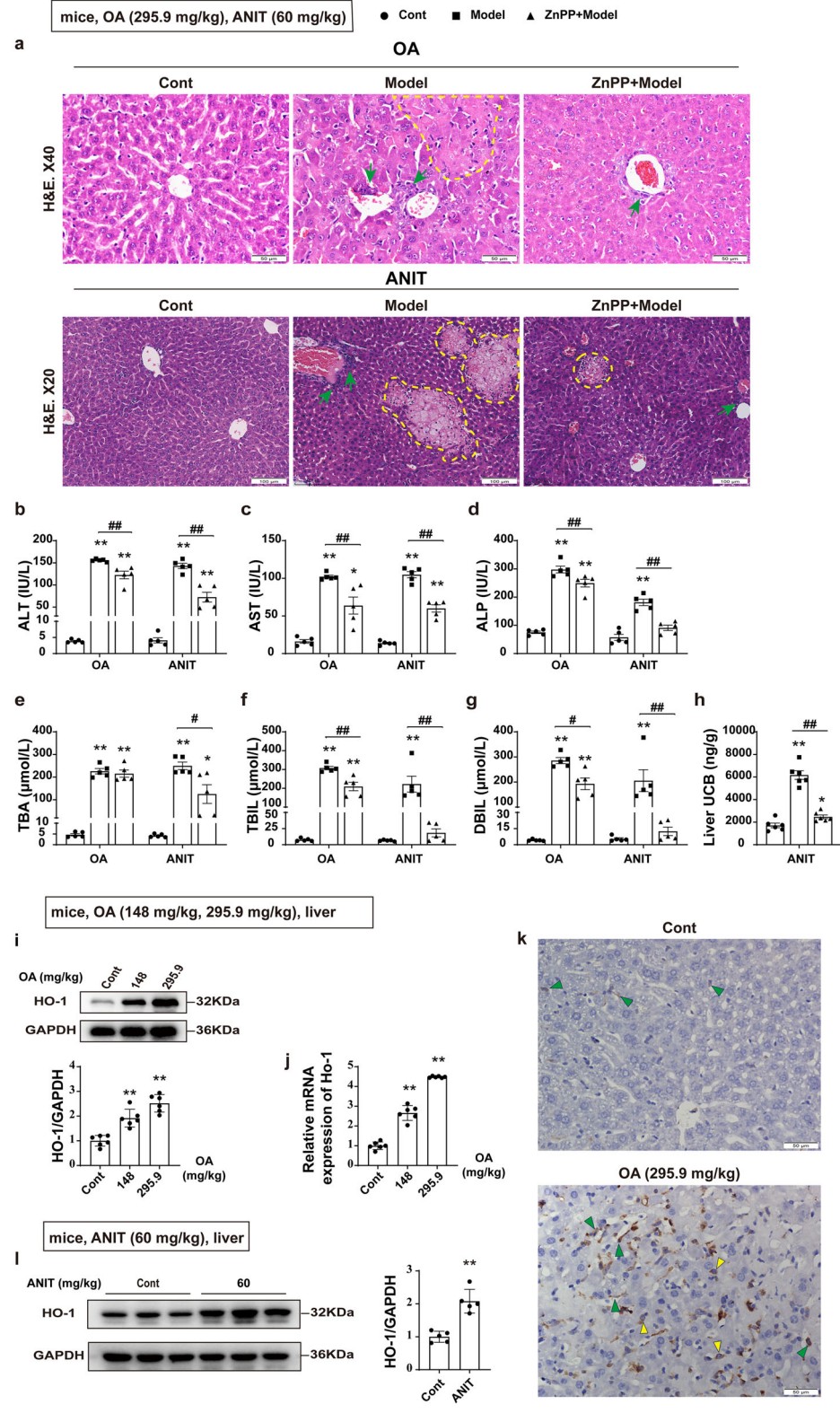

(Supplementary Fig. 5). These results suggest that Nrf2 signaling may play a different (toxic rather than protective) role in the CLI model. To further explore the dual role of Nrf2/HO-1 in liver injury, we determined the expression of HO-1 and found that OA could not induce the increase of HO-1 protein and mRNA expression in *Nrf2*[−/−] mice, and the expression levels of HO-1 protein and mRNA were significantly lower than those in WT mice (Fig. 2m, n). These results indicate that OA can still induce liver injury after Nrf2 is knocked out, but in this case, due to the absence of Nrf2, OA-induced liver damage is not accompanied by the accumulation of bilirubin, thereby leading to a marked reduction in liver injury. Although some antioxidants were downregulated in *Nrf2*[−/−] mice (Supplementary Fig. 6), the overall net effect of Nrf2 knockout was protective against OA-induced CLI due to reduced bilirubin synthesis, revealing a critical role for HO-1 in CLI.

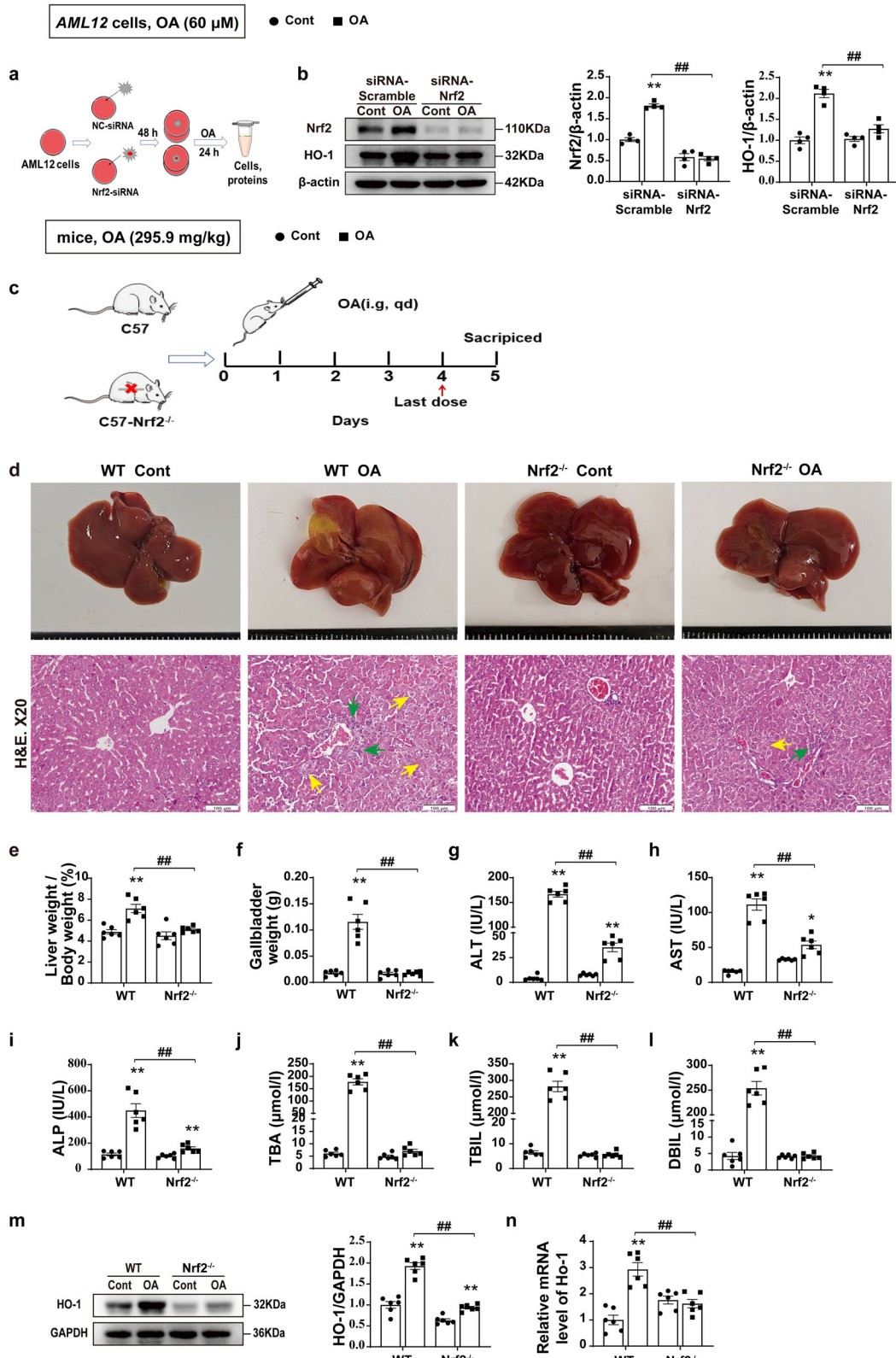

**Fig. 2 | Role of Nrf2-dependent regulation of HO-1 in CLI. a** Schematic illustration of the in vitro experiment to examine the Nrf2-dependent regulation of HO-1 by OA. **b** The protein levels of Nrf2 and HO-1 ($n = 4$). **c** Experimental scheme for OA-induced cholestatic liver injury model in WT and $Nrf2^{-/-}$ mice. **d** Representative H&E staining of liver tissues. Neutrophil infiltration (green arrows) and hepatocyte necrosis (yellow arrows). Scale bar, 100 μm ($n = 6$). **e–l** Gallbladder weight, liver index, and serum levels of ALT, AST, ALP, TBA, TBIL, and DBIL ($n = 6$). **m, n** The mRNA and protein expression levels of HO-1 in the liver ($n = 6$). Student's t test

(**e**, **n**) and nonparametric tests (**b–d** and **f–m**) were used. Data are shown as Mean ± SEM. *$p < 0.05$, **$p < 0.01$, significant difference between the control group and the OA group; #$p < 0.05$, ##$p < 0.01$, significant difference between the WT OA group and the $Nrf2^{-/-}$ OA group. ALP alkaline phosphatase, ALT alanine aminotransferase, AST aspartate aminotransferase, DBIL direct bilirubin, H&E hematoxylin and eosin, HO-1 heme oxygenase-1, OA oleanolic acid, TBA total bile acid, TBIL total bilirubin.

### Eliminating intrahepatic bilirubin accumulation alleviates CLI

The primary role of HO-1 in the cell is to catalyze the catabolism of heme to biliverdin, carbon monoxide, and free ferrous iron. According to reports, reduced iron accumulation plays a role in alleviating liver injury after inhibition of HO-1[19,20], and ferrostatin-1 has been widely used as a ferroptosis inhibitor. Our results found that OA increased iron accumulation (Fig. 3a), but ferrostatin-1 did not improve OA induced liver cell and tissue damage (Fig. 3b), and there were no significant changes in ALT and AST levels (Fig. 3c, d), indicating that inhibition of ferroptosis could not alleviate CLI. To further validate the contribution of HO-1 mediated increase in bilirubin synthesis in CLI, we employed uridine diphosphate glucuronic acid, which converts unconjugated bilirubin into conjugated bilirubin, promoting excretion and reducing the accumulation of bilirubin[30]. Administration of uridine diphosphate glucuronic acid alleviated OA-induced hepatocyte and tissue injury (Fig. 3e), and reduced ALT and AST levels, as well as TBIL, DBIL and indirect bilirubin contents (Fig. 3f–j). These results indicate that excessive accumulation of bilirubin, a metabolite of heme, plays an important role in the process of CLI, while promoting bilirubin excretion can significantly improve CLI.

### Inhibiting hepatobiliary transport leads to bilirubin accumulation and aggravates CLI

Regulation of hepatobiliary transport proteins such as bile salt export pump (BSEP) and multidrug resistance protein (MRP) plays essential role in the maintenance of bile secretion and enterohepatic circulation[31,32]. Studies have shown that MRP2 is the bottleneck of bilirubin excretion, and its loss is sufficient to induce hyperbilirubinemia[33]. We sought to investigate whether modulation of the hepatobiliary transporters had an effect on OA-induced CLI. OA treatment significantly inhibited the protein expression of hepatobiliary transporters in mice and *AML12* cells (e.g., BSEP, MRP2, MRP3, and MRP4) (Fig. 4a, Supplementary Fig. 2). We therefore interrogated whether the use of inhibitor and agonist to manipulate the transporter MRP2 had an impact on bilirubin accumulation and liver injury. The inhibition of MRP2 with its specific inhibitor MK-571 significantly increased ALT, ALP, and bilirubin accumulation (Fig. 4c–e), as well as neutrophil infiltration and exacerbated liver injury (Fig. 4b), while the treatment with sulfanitran (MRP2 agonist) showed a completely opposite effect (Fig. 4f–i). We measured the expression of efflux transporters in the liver of WT mice and *Nrf2*$^{-/-}$ mice to further investigate the role of efflux transporters. The results showed that OA could reduce the mRNA expression of efflux transporters (*Bsep* and *Mrp2*) in WT mice, but there were no significant changes in *Nrf2*$^{-/-}$ mice (Supplementary Fig. 7).

### Excessive accumulation of bilirubin causes mitochondrial impairment

Earlier studies have shown that bilirubin (>50 μM) hinders mitochondrial respiration in isolated liver mitochondria[34]. We speculated that accumulation of high level of bilirubin in the context of cholestasis may directly impair mitochondrial function, worsening the CLI. MRP2 is a bottleneck in bilirubin excretion. In addition, it has been reported that malfunction or blockade of BSEP disrupts normal bile flow and thus adequate bilirubin clearance[35,36]. Therefore, we determined the effect of bilirubin on mitochondria in *AML12* cells and its role in hepatotoxicity, and observed that: (1) cellular injury was induced by bilirubin, and the toxicity increased with increasing concentrations of bilirubin (Fig. 5a); (2) the combination of MRP2 and BSEP inhibitors (MK-571 and BMS, respectively) increased cellular injury caused by bilirubin (Fig. 5b); (3) both bilirubin and OA alone caused hepatotoxicity, while their combined effects resulted in significantly enhanced toxicity (Fig. 5c); and (4) bilirubin significantly inhibited mitochondrial respiration function in the cells measured using Oxygragh-2K (Fig. 5d, e), and increased production of mitochondrial ROS measured using MitoSOX fluorescence probe (Fig. 5f). These results suggest that mitochondrial dysfunction and associated increased oxidative stress may be the culprits of bilirubin-induced cell damage. Next, we determined whether this bilirubin-induced oxidative stress could activate Nrf2-antioxidant signaling.

We detected the protein levels of Nrf2 and HO-1 and observed that bilirubin significantly increased the protein levels of Nrf2 and HO-1, while promoting the transfer of Nrf2 into the nucleus, showing a dose-response relationship (Fig. 5g).

### Inhibition of HO-1 alters mitochondria-related gene expression profile in CLI

To gain more insights on HO-1-dependent mitochondrial dysfunction in the context of CLI, we performed transcriptome sequencing (RNA-seq) analysis on the liver tissues of mice treated with OA. Principal component analysis and hierarchical clustering clearly separated the samples from the control mice and the OA-treated mice into two clusters, while the samples from mice treated with a combination of OA and HO-1 inhibitor (ZnPP) was in the same cluster as the control mice (Fig. 6a). These results suggest that inhibition of HO-1 can significantly reduce liver injury in mice with CLI. GSEA of GO analysis and KEGG analysis showed that changes in mitochondrial- and oxidative phosphorylation-related pathways and genes were among the top of those changes in the liver of OA induced CLI (Fig. 6b, c). Further analysis of gene expression revealed that OA treatment downregulated a group of genes involved in mitochondrial and oxidative phosphorylation pathways, whereas the inhibition of HO-1 with ZnPP effectively offset these changes caused by OA treatment (Fig. 6d, e). These results provide further evidence that HO-1-dependent bilirubin accumulation plays a crucial role in mitochondrial dysfunction and increased mitochondrial ROS production, which contributes to the progression of CLI.

### Expression of Nrf2 and HO-1 is significantly elevated in patients with CLI

Next, we examined the expression levels of *Nrf2* and *HO-1* and related serum biochemical indicators in patients with cholestatic liver disease (Fig. 7 and Supplementary Table 1). Blood samples were collected from 27 normal and 61 patients with CLI. Compared with normal individuals, patients with cholestasis had higher serum ALT, AST, TBA, ALP, and gamma glutamyl trans (Fig. 7f–j), with a significant increase in bilirubin (Fig. 7c–e). Compared with the normal individuals, we observed a significant increase in the expression of *Nrf2* and *HO-1* in the patients' serum (Fig. 7a, b). The significant increase in *HO-1* expression in clinical patients with cholestasis further substantiated our experimental findings that high levels of *HO-1* are detrimental in the context of cholestasis. We proposed that there may exist a feedback loop between Nrf2/HO-1 and the accumulation of bilirubin, where the upregulation of Nrf2-dependent expression of HO-1 leads to an increase in bilirubin accumulation, which in turn induces oxidative stress and activates Nrf2-dependent antioxidant signaling, forming a vicious cycle that promotes the pathological development of CLI (Fig. 7k).

### Mutual regulation between Nrf2/HO-1 activation and bilirubin accumulation

To test the above hypothesis of the mutual regulation between Nrf2/HO-1 and bilirubin accumulation in cholestasis, we conducted subsequent experiments in *AML12* cells. We first observed a robust increase in the protein expression of Nrf2 and HO-1 in cells challenged with bilirubin (Fig. 8a). However, when Nrf2 expression was silenced by siRNA, the increase in HO-1 protein expression induced by bilirubin was dramatically abolished, indicating that bilirubin-induced upregulation of HO-1 is dependent on Nrf2 (Fig. 8a). We then detected mitochondrial ROS generation and found that bilirubin increased mitochondrial ROS production (Fig. 8b). To confirm the role of bilirubin-induced oxidative stress in the Nrf2 mediated regulation of HO-1, we sought to test if antioxidant resveratrol could modulate the interplay between bilirubin and the activation of Nrf2/HO-1. The results showed that while reducing mitochondrial ROS production (Fig. 8b), resveratrol treatment significantly inhibited the expression of Nrf2 and HO-1 protein (Fig. 8c). In addition, resveratrol significantly ameliorated bilirubin induced cytotoxicity (Fig. 8d). These results strongly suggest that the hepatotoxicity of

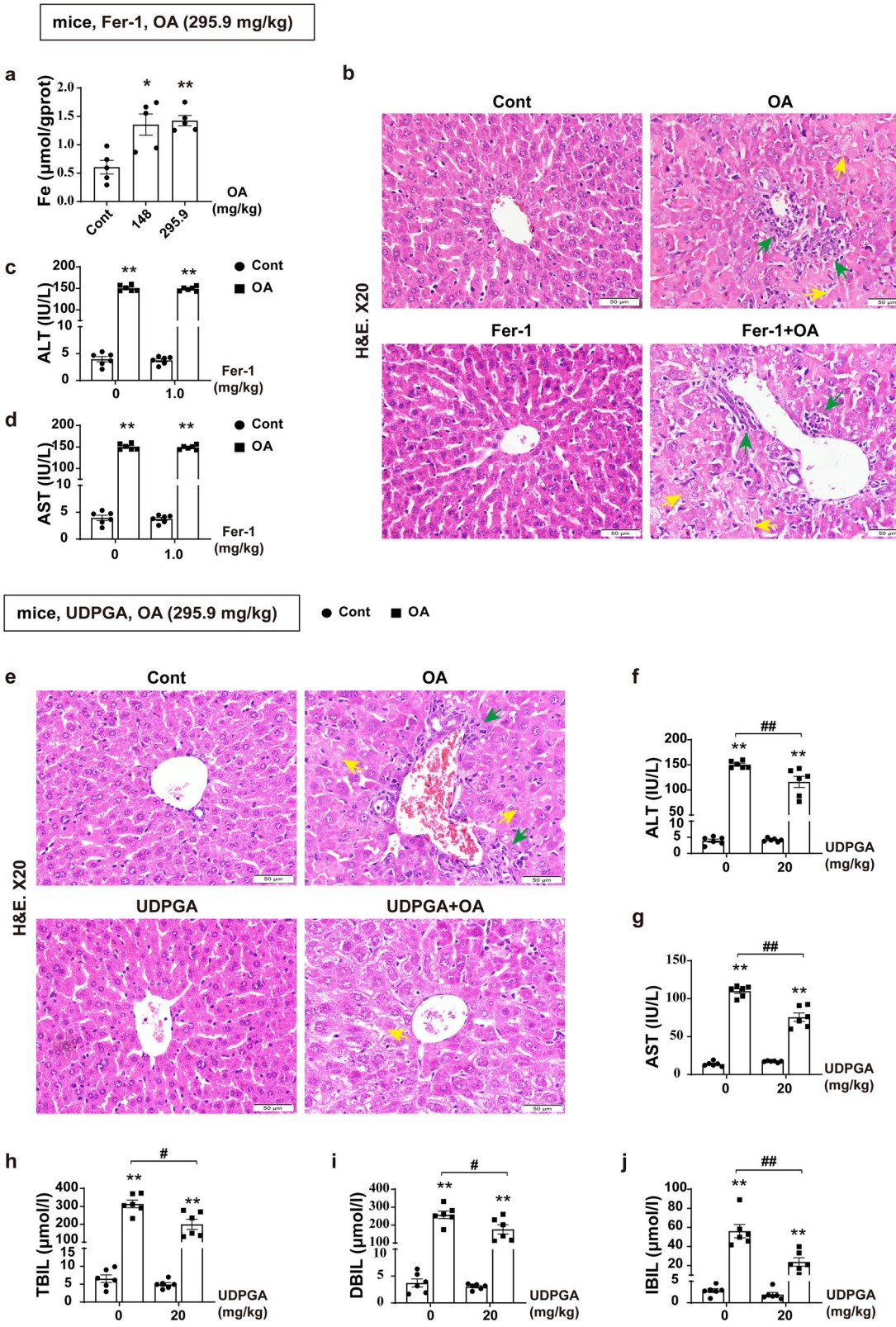

**Fig. 3 | Eliminating intrahepatic bilirubin accumulation alleviates CLI. a** $Fe^{2+}$ contents after administration of WT mice with OA ($n = 5$). **b** Representative H&E staining of liver tissues in mice after administration of 1 mg/kg Fer-1 and 295.9 mg/kg OA. Neutrophil infiltration (green arrows) and hepatocyte necrosis (yellow arrows). Scale bar, 50 μm ($n = 6$). **c, d** Detection of serum ALT and AST ($n = 6$). **e** Representative H&E staining of liver tissues in mice after administration of 20 mg/kg UDPGA and 295.9 mg/kg OA. Scale bar, 50 μm ($n = 5$–6). **f–j** Serum levels of ALT, AST, TBIL, DBIL,

and IBIL ($n = 6$). Nonparametric tests (**c, d** and **f–j**) and one-way ANOVA (**a**) with Games Howell analysis were used. The data are shown as the Mean ± SEM. *$p < 0.05$, **$p < 0.01$, significant difference compared to the control group; #$p < 0.05$, ##$p < 0.01$, significant difference between the OA group and the UDPGA combined OA group. ALT alanine aminotransferase, AST aspartate aminotransferase, DBIL direct bilirubin, Fer-1 ferrostatin-1, H&E hematoxylin and eosin, IBIL indirect bilirubin, OA oleanolic acid, TBIL total bilirubin, UDPGA uridine diphosphate glucuronic acid.

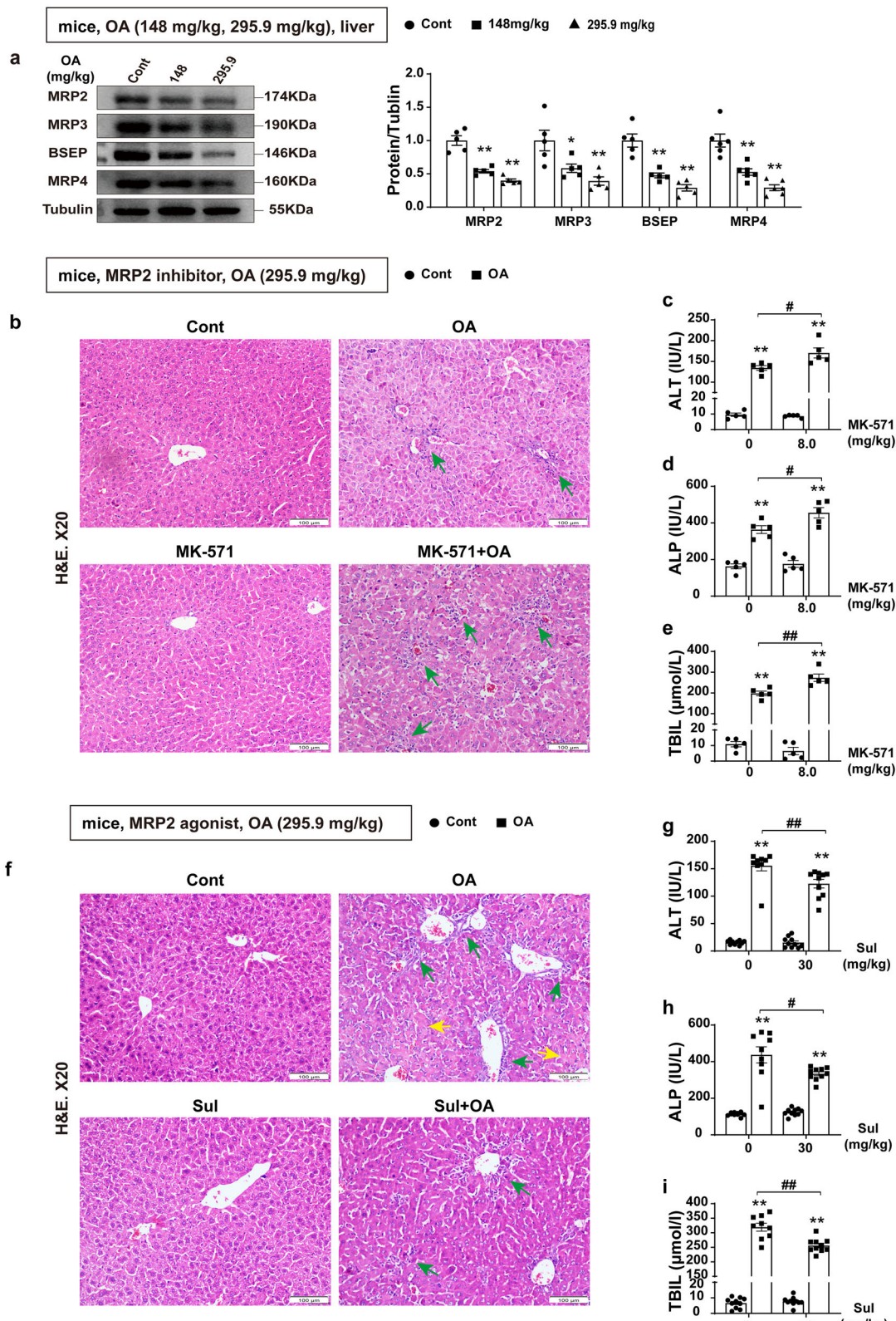

**Fig. 4 | Inhibiting hepatobiliary transport leads to bilirubin accumulation and aggravates cholestatic liver injury. a** The protein levels of BSEP, MRP2, MRP3, and MRP4 in mice after administration of 148 and 295.9 mg/kg OA (*n* = 5–6). **b–e** WT male mice were pre-treated with MK-571 followed by OA for 4 days (*n* = 5). **b** Representative H&E staining of liver tissues (scale bar, 100 μm). Neutrophil infiltration (green arrows). **c–e** Serum levels of ALT, ALP, and TBIL. **f–i** WT male mice were pre-treated with Sul followed by OA for 4 days (*n* = 9–10). **f** Representative H&E staining of liver tissues (scale bar, 100 μm). Neutrophil

infiltration (green arrows) and hepatocyte necrosis (yellow arrows). **g–i** Serum ALT, ALP and TBIL. One-way ANOVA (**a**) with Tukey's post hoc test or Games Howell, Student's t test (**d**) and nonparametric tests (**c**, **e** and **g–i**) were used. The data are shown as Mean ± SEM. *$p < 0.05$, **$p < 0.01$, significant difference compared to the control group; #$p < 0.05$, ##$p < 0.01$, significant difference between the OA group and the MK-571 or Sul combined OA group. ALP alkaline phosphatase, ALT alanine aminotransferase, BSEP bile salt export pump, H&E hematoxylin and eosin, MRP multidrug resistance protein, OA oleanolic acid, Sul sulfanitran, TBIL total bilirubin.

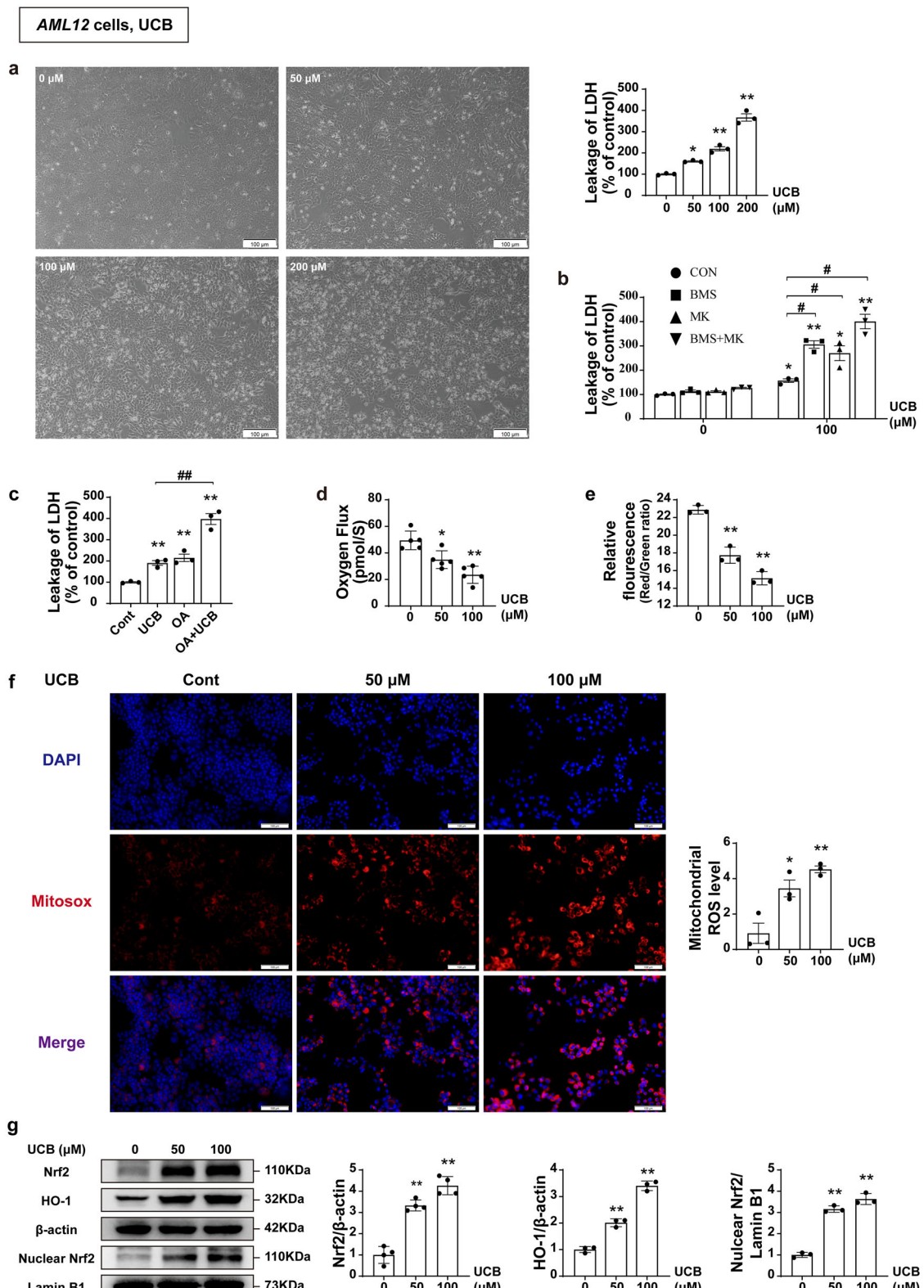

**Fig. 5 | Excessive accumulation of bilirubin in CLI causes mitochondrial impairment. a** Cell morphology and detection of LDH leakage after *AML12* cells were challenged with UCB for 12 h ($n = 3$), scale bar, 100 μm. **b** Detection of LDH leakage. *AML12* cells were pretreated with MRP2 inhibitor MK-571 (20 μM) and BSEP inhibitor BMS (1 μM) and then exposed to UCB (100 μM) for 12 h ($n = 3$). **c** Effect of combination of OA and UCB on cell survival. *AML12* cells were pretreated with OA (60 μM) and followed by the treatment with UCB (100 μM) for 12 h ($n = 3$). **d**, **e** Oxygen flux and mitochondrial membrane potential ($n = 5$). **f** MitoSOX Red

staining, scale bar, 100 μm. The box plot displays the signal strength of the staining ($n = 3$). **g** The protein levels of Nrf2, HO-1, and nuclear Nrf2 in the different groups after UCB challenge ($n = 3–4$). One-way ANOVA (**a**, **d**–**g**) with Tukey's post hoc test and Student's t test (**b**, **c**) were used. All data are shown as Mean ± SEM. *$p < 0.05$, **$p < 0.01$; #$p < 0.05$, ##$p < 0.01$. HO-1 heme oxygenase-1, LDH lactate dehydrogenase, Nrf2 nuclear factor erythroid 2-related factor 2, OA oleanolic acid, UCB unconjugated bilirubin.

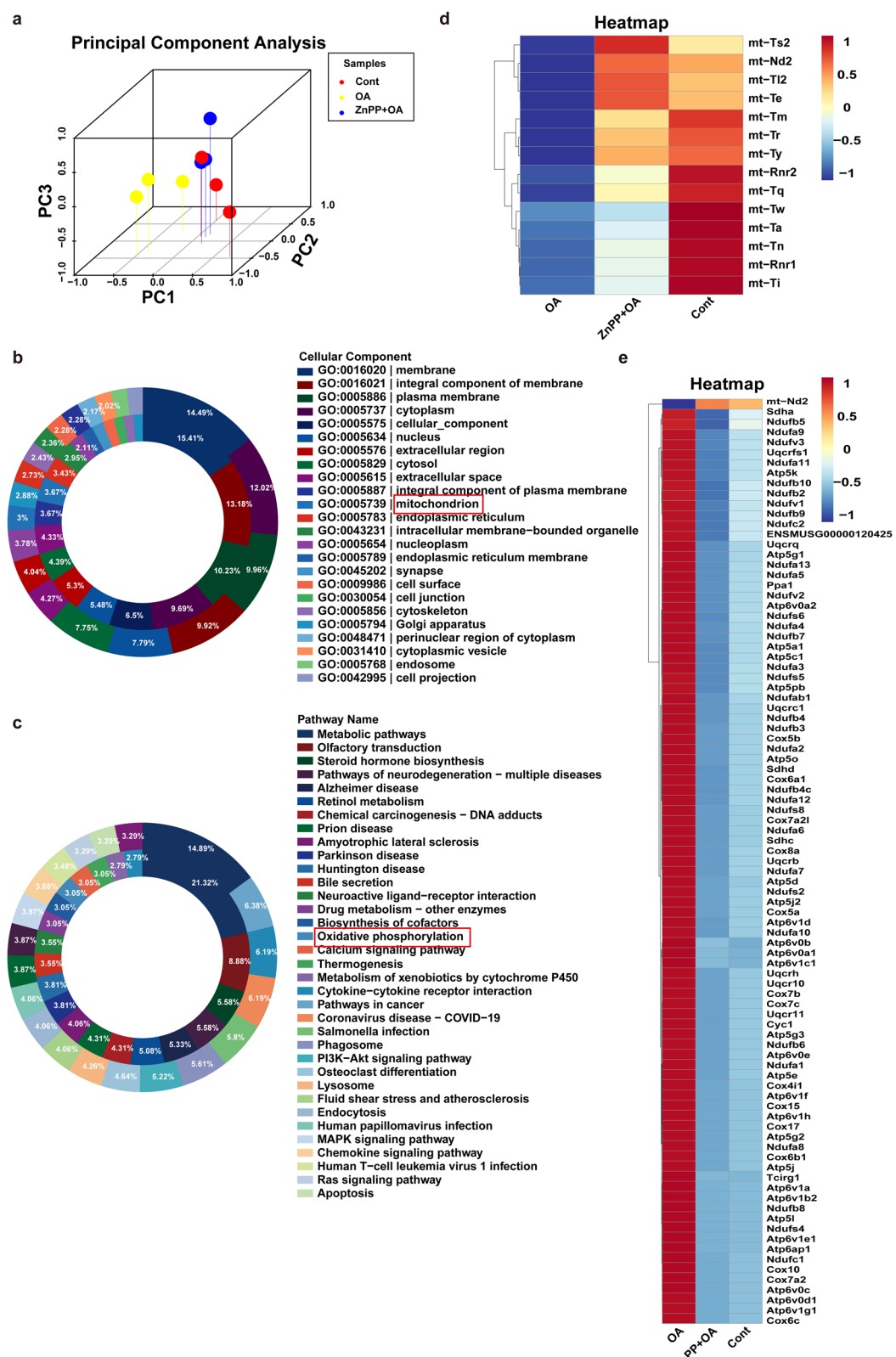

**Fig. 6 | Inhibition of HO-1 alters mitochondria-related gene expression profile in CLI. a** Principal Component Analysis (PCA) of RNA-seq data from mice fed a control, OA, and OA combined ZnPP diet for 4 days (*n* = 3). **b, c** Gene Ontology (GO) enrichment analysis and Kyoto Encyclopedia of Genes and Genomes (KEGG) pathway enrichment analysis (control vs OA). The outer circle indicates the upregulation and the inner circle indicates the downregulation. **d, e** Heatmap of mitochondrial-related and oxidative phosphorylation-related genes. Red indicates the upregulation, and blue indicates the downregulation in heatmaps. Data set. *n* = 3 per group. RNA-seq, RNA sequencing.

**Fig. 7 | Expression of Nrf2 and HO-1 is significantly elevated in patients with CLI.** Serum samples were obtained from 27 normal individuals and 61 CLI patients. **a, b** HO-1 and Nrf2 mRNA expression. **c–e** Serum levels of TBIL, DBIL, and indirect bilirubin (IBIL). **f–j** Serum levels of ALT, AST, ALP, GGT, and TBA. **k** Schematic illustration on how bilirubin may regulate Nrf2/HO-1 pathway-mediated oxidative stress during cholestatic liver injury. Nonparametric statistical test (**a–j**) was used. All data are presented as the mean ± SEM. ALP alkaline phosphatase, ALT alanine aminotransferase, ARE antioxidant responsive element, AST aspartate aminotransferase, BLVRA biliverdin reductase A, DBIL direct bilirubin, GCLC glutamate-cysteine ligase catalytic subunit, GGT gamma Glutamyl Trans, HO-1 heme oxygenase-1, IBIL indirect bilirubin; NQO1 NAD(P)H-quinone oxidoreductase 1, Nrf2 nuclear factor erythroid 2-related factor 2, ROS reactive oxygen species, TBA total bile acid, TBIL total bilirubin.

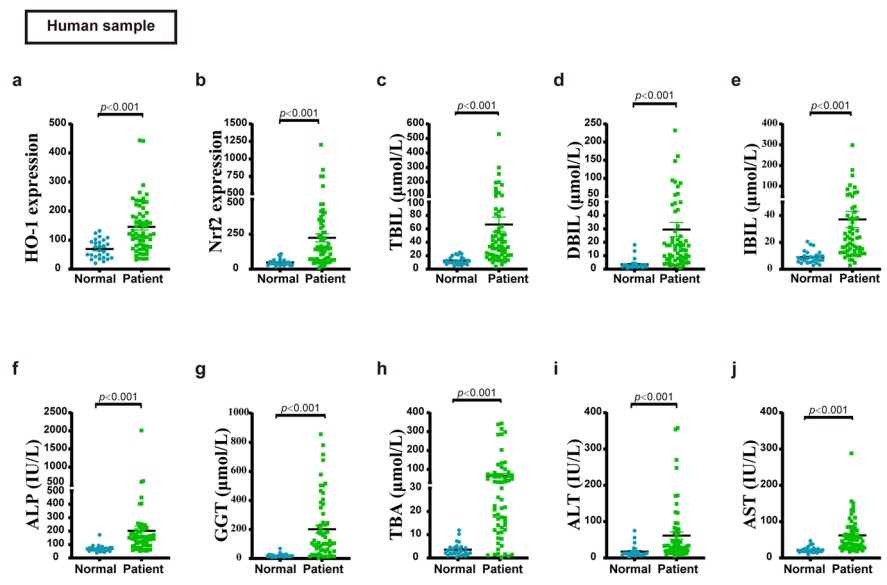

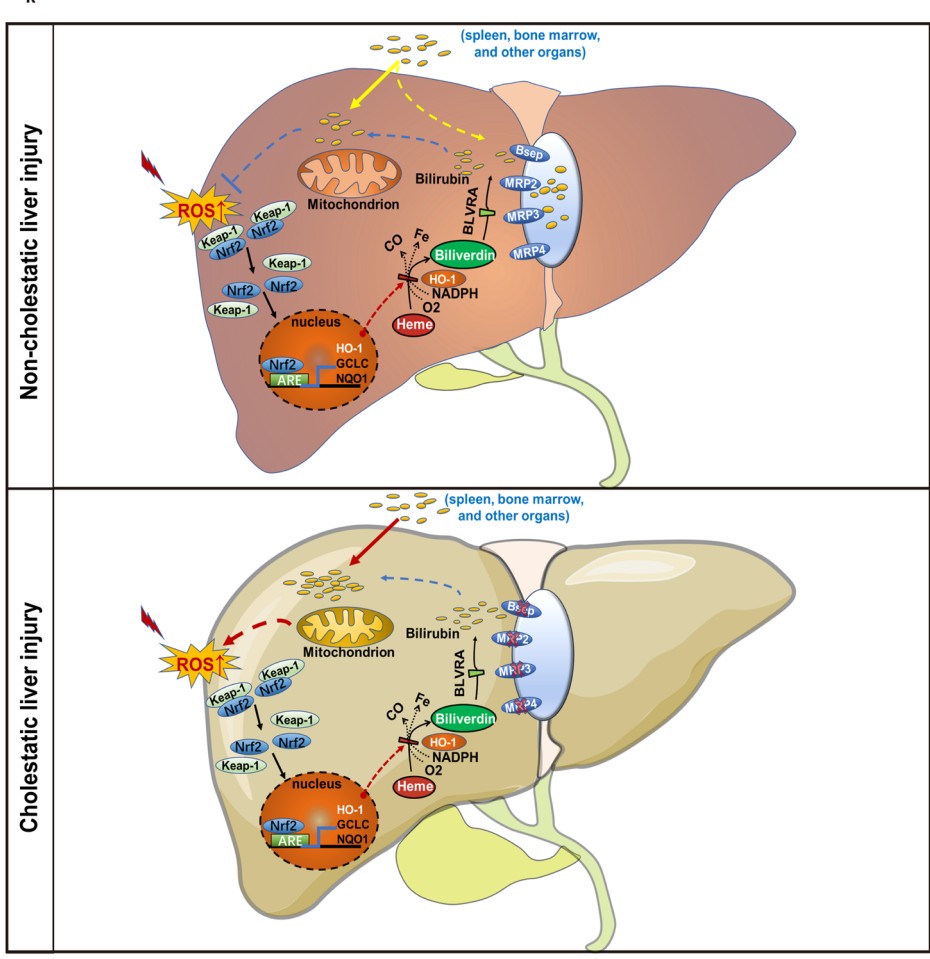

bilirubin is related to oxidative damage. To confirm the key role of mitochondrial ROS in the bilirubin induced cytotoxicity, we used lentivirus HBLV-SOD2 to infect *AML12* cells and established a cell line of mitochondrial-targeted SOD2 overexpression (Supplementary Fig. 8) to further explore the hepatotoxicity of bilirubin. Overexpression of SOD2 profoundly inhibited the expression of Nrf2 and HO-1 protein induced by bilirubin (Fig. 8e), and significantly reduced mitochondrial ROS production (Fig. 8f). These results substantially support our hypothesis that excessive accumulation of bilirubin in the liver leads to impaired mitochondrial function and increases mitochondrial ROS production, thereby inducing oxidative stress and activating Nrf2/HO-1 antioxidant signaling, leading to increased expression of HO-1 protein and more

**Fig. 8 | Bilirubin induced upregulation of Nrf2/HO-1 is dependent on mitochondrial ROS.** **a** Silencing of Nrf2 in *AML12* cells with siRNA. The protein levels of Nrf2 and HO-1 in the different groups after UCB challenge ($n = 4$). **b–d** *AML12* cells were pretreated with Res (10 µM), followed by exposure to UCB (100 µM) for 12 h. Measurement of LDH leakage ($n = 3$), protein expression levels of Nrf2 and HO-1 ($n = 4$), and mitochondrial ROS production (scale bar, 100 µm) ($n = 3$) after treatment with UCB (100 µM). The box plot displays the signal strength of the staining ($n = 3$). **e** Effect of mitochondrial-targeted SOD2 overexpression on protein expression levels of Nrf2 and HO-1 ($n = 4$). **f** Mitochondrial ROS production (scale bar, 100 µm). Overexpression of SOD2 by Lentivirus in *AML12* cells, followed by UCB (100 µM) for 12 h ($n = 3$) (**e, f**). Student's t test (**a–c** and **e**) and non-parametric tests (**e, f**) were used. The data are shown as the Mean ± SEM. *$p < 0.05$, **$p < 0.01$, significant difference compared to the control group; #$p < 0.05$, ##$p < 0.01$. HO-1, heme oxygenase-1; LDH, lactate dehydrogenase; Nrf2, nuclear factor erythroid 2-related factor 2; Res, resveratrol; SOD2, superoxide dismutase 2; UCB, unconjugated bilirubin.

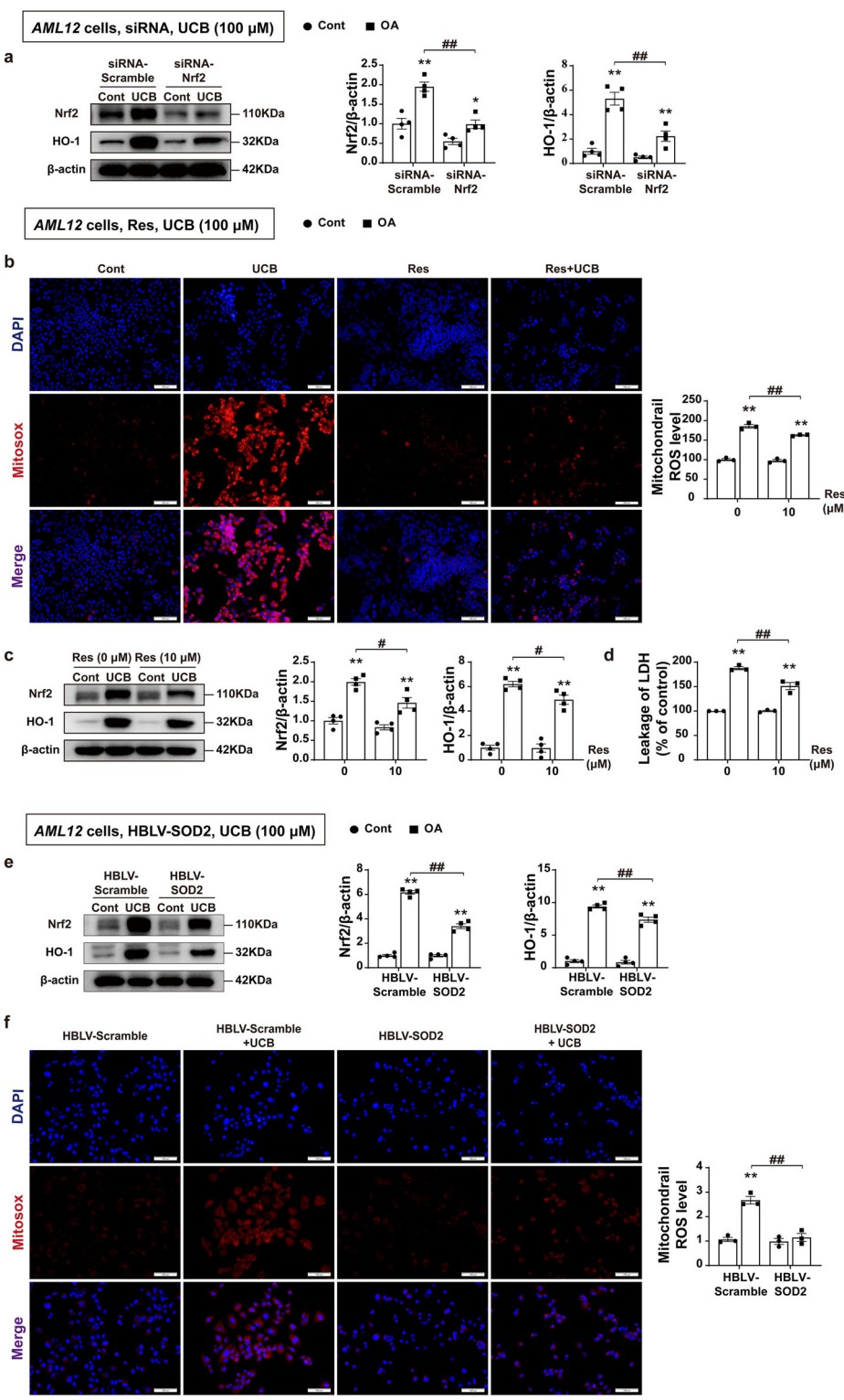

bilirubin synthesis, and ultimately forming a vicious cycle that exacerbates CLI.

## Discussion

The transcription factor Nrf2 plays a crucial role in regulating the expression of multiple detoxifications and antioxidant defense genes and has been widely involved in the protective mechanisms of various liver diseases[37]. However, the activation of Nrf2 may not always have a beneficial effect on liver protection. In the present study, we demonstrate that the activation of

Nrf2 antioxidant signaling is detrimental in the context of cholestatic liver injury. Moreover, we show that the knockout of Nrf2 does not increase liver damage, but rather reduces liver damage in chemically induced CLI in vivo.

A major finding of this study is that Nrf2 activation aggravates cholestatic liver injury, whereas it is protective against liver injury in non-CLI. The activation of Nrf2 upregulates HO-1, leading to more synthesis and accumulation of bilirubin in the liver, which in turn enhances liver injury in the context of CLI. Although the activation of Nrf2 may also upregulate a battery of antioxidants that have protective effect against oxidative stress, the

net effect of Nrf2 upregulation is to exacerbate liver injury due to the toxicity of bilirubin accumulation in the liver. CLI, especially drug (including herbal medicine) induced CLI, causes increased oxidative stress, which induces Nrf2 activation[13,15,38]. Under unstressed conditions, Nrf2 binds to Keap1 in the cytoplasm and always maintains at low levels. In response to oxidative stress, Nrf2 translocates into the nucleus and induces the expression of a group of downstream antioxidant genes, such as NAD(P)H-quinone oxidoreductase 1, SOD2, glutamate-cysteine ligase catalytic subunit, HO-1 and glutathione peroxidase, etc. Among them, HO-1 is involved in the synthesis of bilirubin. The role of Nrf2/HO-1 mediated increase in bilirubin synthesis in CLI has not been described yet. In this study, we not only demonstrated that the Nrf2/HO-1 activation-mediated bilirubin accumulation is hepatotoxic in CLI, but also showed that the inhibition of HO-1 or Nrf2 knockout conferred a hepatoprotective benefit in the setting of CLI. On the contrary, $Nrf2^{-/-}$ mice exhibited increased liver damage in the CCl$_4$ induced non-CLI model.

Our results revealed that bilirubin accumulation is a critical player in exacerbation of cholestatic liver injury mediated by Nrf2 activation, which is related to the regulation of export transporters. In fact, cholestasis, especially chemical induced cholestasis, usually leads to impaired bilirubin excretion[39]. Impairment/downregulation of export transporters (such as MRP2, MRP3, MRP4, BSEP, etc) has been frequently involved in the mechanism of bile acids accumulation in the liver and implicated in various drug-induced cholestatic liver injury[40]. Previous studies show that the regulation of export transporters and the activation of MRP2 are involved in bilirubin metabolism and reduce liver toxicity[41]. Consistent with this, we demonstrated that the activation and inhibition of MRP2 alleviate or worsen cholestatic liver injury in vivo, respectively. Importantly, in this study we revealed a direct connection between Nrf2/HO-1 signaling pathway and bilirubin accumulation as well as cholestatic liver injury. However, in the non-CLI model induced by CCl$_4$, the inhibition of Nrf2 and HO-1 expression increased hepatotoxicity, showing a significantly opposite effect to the CLI model. These results further indicate that the harmful effects of Nrf2/HO-1 signaling-dependent bilirubin accumulation are unique to the course of cholestasis, which is the main finding of this study. When cholestasis occurs, bilirubin synthesis is enhanced due to the activation of Nrf2/HO-1 induced by oxidative stress, and on the other hand, the outflow of bilirubin is inhibited, thereby leading to the excessive accumulation of bilirubin and liver injury. However, the dysregulation of exporter transporters (BSEP and MRP2) may also affect the accumulation of bile acids, one of the important players in the pathology of CLI[42]. Indeed, we found that *Bsep* and *Mrp2* were downregulated in the liver of CLI mice in a Nrf2-dependent manner (Fig. S7). Although the mechanism of Nrf2-mediated downregulation of *Bsep* and *Mrp2* has yet to be further investigated, these results suggest a role of bile acid regulation in the pathogenesis of liver injury, which is entangled in the exacerbation of liver injury induced by Nrf2/HO-1 activation in CLI. In this regard, further dissecting the molecular mechanism of Nrf2-mediated regulation of BSEP and MRP2 and its role in bilirubin accumulation may be necessary for comprehensively understanding the mechanism by which activation of Nrf2/HO-1 signaling pathway aggravates cholestatic liver injury.

Previous studies have demonstrated that bilirubin has antioxidant potential or acts as a powerful signaling molecule, but these potentially beneficial effects are based on "physiological concentrations" or "slightly elevated" levels of bilirubin. The dangers of hyperbilirubinemia are well known, but the specific mechanism of toxic action of bilirubin remains unclear. Earlier studies have shown that high concentrations of bilirubin can inhibit the mitochondrial function of isolated liver mitochondria[34]. Here, we clearly demonstrated that bilirubin treatment induced mitochondrial respiratory dysfunction and cellular injury, and the cytotoxicity of bilirubin was significantly enhanced after the inhibition of BSEP and MRP2, which was consistent with the harm of bilirubin in cholestasis observed in vivo. Importantly, we found that the significant increase in mitochondrial ROS production was associated with bilirubin treatment. We postulated that this bilirubin induced mitochondrial ROS may in turn activate the Nrf2/HO-

1 signaling, leading to more bilirubin accumulation and a vicious cycle. Indeed, our results revealed that bilirubin activates Nrf2 and HO-1 protein expression and induces the translocation of Nrf2 into the nucleus to trigger the antioxidant mechanism. Therefore, we propose that the mitochondrial dysfunction and mitochondrial ROS generation may play a central role in the mechanism of bilirubin induced hepatotoxicity in the context of cholestatic liver injury. In support of this hypothesis, mitochondrial-targeted overexpression of SOD2 not only reduced mitochondrial ROS generation, but also significantly ameliorated cellular injury induced by bilirubin. Additionally, inhibition of HO-1 led to restoration of abnormal mitochondria-related gene expression and improved mitochondrial function and oxidative phosphorylation levels. Despite the impact of bilirubin on mitochondrial function, many questions have yet to be defined, such as what are the mitochondrial target(s) of bilirubin or the specific mode of action of bilirubin on mitochondria under cholestatic conditions. In this regard, further characterization of mitochondrial bilirubin may be valuable, given the excessive accumulation of bilirubin in liver tissue in the setting of CLI and the direct effects of bilirubin on isolated mitochondria[34]. Although the exact molecular events linking bilirubin to mitochondrial dysfunction warrant further studies, our results in this study collectively reveal the key role of mitochondrial impairment in bilirubin induced hepatotoxicity. It is worth noting that emerging evidence suggests that HO-1 may induce ferroptosis through iron accumulation or other unknown mechanisms, which is related to liver damage and neurodegeneration[43,44]. Excessive iron produces ROS through Fenton reaction, resulting in ferroptosis[45]. In this study, although OA induced iron accumulation in the liver, the intervention of iron inhibitor ferrostatin-1 on iron accumulation did not reduce the liver injury, indicating that the beneficial effect of inhibition of HO-1 with its antagonist on OA induced cholestatic liver injury can be attributed to a decrease in bilirubin accumulation, but is not related to changes in iron content in the liver.

However, we do not refute the antioxidant effect of Nrf2's upregulation of antioxidant signaling. In fact, oxidative stress has been widely believed to be related to the pathogenesis of cholestasis, although the exact mechanism remains to be fully elucidated[46]. Nrf2 knockout did result in downregulation of antioxidants NAD(P)H-quinone oxidoreductase 1, glutamate-cysteine ligase catalytic subunit and SOD2, which may compromise the antioxidant capacity to some extent. Nevertheless, in the case of cholestasis, the net effect of activation of Nrf2 is to exacerbate liver damage, which is due to increased synthesis and clearance disorders leading to excessive accumulation of bilirubin. This seems to contradict the previous results showing that sustained Nrf2 activation has a hepatoprotective effect on cholestasis-related liver injury[47]. However, our model (OA or ANIT) is chemically induced cholestatic liver injury, which differs from the mechanism of action of bile duct ligation. In addition, there are studies demonstrating that the overexpression of HO-1 accelerates the progression of chronic cholestasis to liver fibrosis[19,20]. Fully elucidating the role of Nrf2/HO-1/bilirubin axis in these situations may help to further understand the dual function of activating Nrf2, that is, Nrf2 activation is harmful to cholestatic liver injury, but has a protective effect on non-cholestatic liver injury.

Importantly, in patients with cholestatic liver disease, the expression of *Nrf2* and *HO-1* in the serum is significantly increased. Particularly, the expression level is directly correlated with the serum content of total, direct, and indirect bilirubin in the patients. These results strongly suggest that the Nrf2/HO-1/bilirubin axis may play a crucial role in the development of cholestatic liver injury, and that the intervention of Nrf2/HO-1 may be a potential therapeutic target for the treatment of cholestatic liver disease. In addition, many liver protection agents, including some Chinese herbal medicines and their effective components, exhibit their hepatoprotective effects through the activation of Nrf2-dependent antioxidant signaling[48,49]. Our findings in this study suggest that when using these Nrf2 activators as hepatoprotective agents in the context of cholestatic liver disease, special attention should be paid to liver toxicity, because on the contrary, Nrf2 activation may lead to more severe liver damage.

In conclusion, our findings demonstrate that activation of Nrf2 exacerbates liver damage in the context of cholestasis by upregulating the expression of HO-1. Our study results unravel that activation of Nrf2/HO-1 results in excessive accumulation of bilirubin in the liver, thereby damaging mitochondrial homeostasis and inducing oxidative stress, leading to Nrf2 activation, forming a vicious cycle, and exacerbating liver damage in cholestatic liver disease. Our findings suggest that the intervention of Nrf2/HO-1 signaling has a potential translational value in the clinical treatment of cholestatic liver disease.

## Methods

### Human samples

Blood samples from patients with CLI ($n = 61$) and blood samples from normal individuals ($n = 27$) were collected from May 2022 to December 2022 in the Department of Gastroenterology of the Affiliated Hospital of Zunyi Medical University (Guizhou, China). The pathological diagnosis was made by gastroenterologists of the Affiliated Hospital of Zunyi Medical University based on the patient's clinical manifestations, signs, biochemical indicators, imaging examination and gastroenteroscopy. Clinical information (sex, age, and pathological diagnosis) was provided by the Affiliated Hospital of Zunyi Medical University. Written consent to participate in this study was obtained from all patients. The study protocol for cholestatic liver injury was approved by the Ethical Review Committee of Affiliated Hospital of Zunyi Medical University (No. KLL-2023-029). The authors declare that this study adhered to the Declarations of Helsinki and Istanbul. All ethical regulations relevant to human research participants were followed.

### Experimental animals

Male *C57BL/6J* (wild-type, WT) mice aged 6-8 weeks were purchased from the SPF Biotechnology Co., Ltd (Beijing, China). *Nrf2$^{-/-}$* mice were provided by Dr. Lili Ji, Institute of Chinese Medicine, Shanghai University of Traditional Chinese Medicine[28,50]. For all experiments, *Nrf2$^{-/-}$* and WT mice were male mice with male *C57BL/6J* background, aged 6-8 weeks. And all *Nrf2$^{-/-}$* mice were genotyped using PCR with mouse tail DNA, and *Nrf2* gene fragment was amplified using the forward primer 5′-TGAG ATCTGCCTTCTTCTTGCC-3′, and reverse primer 5′-CACGAGA GTGTACCTGGGAGTAGC-3′. The genotyping was performed by BGI Gene Technology Co., Ltd. The *Nrf2* knockout allele sequence had two missing base pairs (GA) between 340 and 350. Thereafter, mice were bred at the SPF grade animal house of Key Laboratory of Basic Pharmacology of Ministry of Education, Zunyi Medical University. All animals were kept under temperature-controlled (22–25 °C) conditions. All animal experimental protocols in the present study were operated according to the Guide for the Care and Use of Laboratory Animals published by the United States National Institutes of Health (NIH Publication No. 85-23, revised 1996) and were approved by the Experimental Animal Ethics Committee of the Zunyi Medical University (No. ZMU 21-2203-524). And we have complied with all relevant ethical regulations for animal use. All mice were housed in recyclable individually ventilated cages in Zunyi Medical University animal facility on a 12 h light/dark cycle. Mice were fed a normal chow diet (protein, 20.6%; fat, 12.0%; carbohydrates, 67.4%; 1010088; Jiangsu Xietong, China) and sterile water, except for fasting period. All interventions were done during the light cycle.

### Animal models of OA-induced cholestatic liver injury

WT mice and *Nrf2$^{-/-}$* mice aged 6 weeks were randomly assigned to 4 groups: WT littermate control group, OA induced WT group, *Nrf2$^{-/-}$* littermate control group, OA induced *Nrf2$^{-/-}$* group. OA induced-WT group and -*Nrf2$^{-/-}$* group were constructed by gavage with OA (295.92 mg/kg, Sigma-Aldrich, St. Louis, MO, USA) for 4 days[51,52], whereas the other two groups received the same volume of corn oil intragastrically.

To observe the effects of HO-1, iron and bilirubin on OA-induced cholestatic liver injury, (1) protoporphyrin IX zinc (ZnPP) (282820, Sigma-Aldrich, St. Louis, MO, USA), an inhibitor of HO-1, was intraperitoneally administrated at dose of 20 mg/kg or equal volume of 0.9% normal saline at 2 h prior to corn oil or OA gavage; (2) ferrostatin-1 (Fer-1, 1 mg/kg) (SML0583, Sigma-Aldrich, St. Louis, MO, USA), an iron apoptosis inhibitor, was intraperitoneally administered at dose of 1 mg/kg or equal volume of vehicle DMSO at 2 h prior to corn oil or OA gavage. Fer-1 was dissolved in DMSO at 200 mg/kg and was eventually diluted with saline to 0.5% DMSO (1 mg/kg) (0.5% DMSO diluted with 0.9% normal saline for control groups) as previously described[53]; (3) uridine diphosphate glucuronic acid (UDPGA, 20 mg/kg) (U5625, Sigma-Aldrich, St. Louis, MO, USA), a bilirubin bound glucuronic acid donor, was intraperitoneally administered at dose of 20 mg/kg or equal volume of 0.9% normal saline at 2 h prior to corn oil or OA gavage; (4) MK-571 (8 mg/kg) (115103-85-0, Med Chem Express, NJ, USA), an inhibitor of MRP2, was intraperitoneally administered at dose of 8 mg/kg or equal volume of PBS at 2 h prior to corn oil or OA gavage; (5) sulfanitran (Sul, 30 mg/kg) (122-16-7, Med Chem Express, NJ, USA), an agonist of MRP2, was intraperitoneally administered at dose of 30 mg/kg or equal volume of vehicle DMSO at 2 h prior to corn oil or OA gavage. All animals were fasted 12 h before execution. After successful modeling, blood and liver tissues were collected for subsequent analysis.

### Animal models of ANIT-induced cholestatic liver injury

ANIT-induced cholestatic liver injury was established by intragastric administration of α-naphthylisothiocyanate (ANIT) (551-06-4, Aladdin, Beijing, China) to male mice aged 6 weeks as previously described[54]. The ZnPP (20 mg/kg) was given to mice by intraperitoneal injection for consecutive 2 days once a day. Control and ANIT group were administrated normal saline each day accordingly. Except for the control group, other groups were treated with 60 mg/kg ANIT (dissolved in corn oil) after ZnPP injection on the first day, while the mice in the control group received intragastrical treatment with corn oil. All animals were sacrificed to collect serum and liver tissue at 48 h after ANIT treatment. All animals were fasted 12 h before execution. After successful modeling, blood and liver tissues were collected for subsequent analysis.

### Animal models of non-cholestatic liver injury

To model non-CLI, male mice aged 6 weeks were intraperitoneally injected with carbon tetrachloride ($CCl_4$). The $CCl_4$ (R051866, Rhawn, Shanghai, China) induced liver injury model was established as previously described[28,55]. To define the role of HO-1 in the non-CLI model, HO-1 inhibitor (ZnPP) was administered before being injected with $CCl_4$ (10 μL/kg and 20 μL/kg, diluted in corn oil), and the mice in the control group were intraperitoneally injected with corn oil. To further determine the role of Nrf2 in non-CLI models, both WT mice and *Nrf2$^{-/-}$* mice were also intraperitoneally injected with $CCl_4$ for 4 days. All animals were fasted 12 h before execution. After successful modeling, blood and liver tissues were collected for subsequent analysis.

### RNA-seq and data processing

Total RNA was extracted using Trizol reagent (Cat #15596018, Thermo Fisher). The integrity and purity of total RNA were evaluated using Bioanalyzer 2100 and RNA 6000 Nano LabChip Kit (Cat #5067-1511, Agilent, CA, USA). A sequencing library was constructed using samples with RIN numbers greater than 7. Part of the specific analysis methods is described in the supplementary file. RNA quality testing, genome-wide library construction, sequencing, and result analysis were completed by Heyuan Biotechnology Co., Ltd. (Shanghai, China).

### Principal component analysis (PCA)

The principal component analysis was performed to analyze and reveal the structure/relationship of the samples based on data from RNA-seq by princomp function in R.

### Gene set enrichment analysis

Gene set enrichment analysis was performed using gene sets of GO terms to rank genes based on their expression change and to analyze the whole-

genome changes. Gene sets with $p$ values < 0.05 and false discovery rate (FDR) values < 0.25 were considered statistically significant. The outer circle contains upregulated genes, and the inner circle contains downregulated genes.

## KEGG pathway enrichment analysis
A KEGG pathway enrichment analysis was performed to analyze the differential expression genes through GSEA software (version 4.1.0). KEGG pathway annotations were downloaded from the KEGG database, and $p$ values < 0.05 were considered statistically significant enrichment pathways.

## LC-MS quantification of liver UCB concentration
UCB concentrations were quantified using Liquid chromatography-mass spectrometry (LC-MS) (UltiMate 3000 RS, Q Exactive high-resolution mass spectrometer, Thermo, MA, USA). The sample was mixed with pure methanol, and the mixture was ground and centrifuged. After the final centrifugation steps, the supernatant was collected and filtered through a membrane using a 0.22 μ filter, and the filtrate was taken for analysis (suitable for LC-MS analysis). For LC separation, a Venusil C18 Plus (5 μm, $2.1 \times 50$ mm, Agela, DE, USA) was used with a flow rate of 0.4 mL/min at 40 °C. The injection volume was 5 μL. The mobile-phase solvents were 0.1% formic acid in water (A) and methanol (B). The gradient elution conditions were set at 20% B, linearly increased to 98% B during the next 2 min (min 0–2), and 98% B for 4 min (min 2–6), and finally linearly decreased to 20% B during the next 4.10 min (min 6–10.10), and 20% B lasted for 5 min (min 10.10–15.10). The electrospray ionization conditions for MS were set as follows: positive ion source; ion source voltage, 4000 V; collision gas, 50Arb; and auxiliary gas, 15Arb. The chromatogram collection and integration of compounds were processed using Xcalibur 4.0 (Thermo, MA, USA) software, and linear regression was performed with $1/x^2$ as the weighted coefficient.

## Measurements of liver injury
Serum concentrations of alanine aminotransferase (ALT), aspartate aminotransferase (AST), total bilirubin (TBIL), direct bilirubin (DBIL), alkaline phosphatase (ALP) and total bile acid (TBA) were measured using a microplate reader (Multiskan GO, Thermo Fisher Scientific, Waltham, MA), according to the manufacturer's instructions.

## Iron assay
The iron levels in liver were determined using the Iron Assay Kit from Elabscience Biotechnology Co., Ltd (Wuhan, China) according to the manufacturer's instructions. In brief, liver was rapidly homogenized in Extracting Solution and centrifuge at 12,000 ×$g$ for 10 min at 4 °C to remove insoluble material. Approximately 300 μL samples were added to 150 μL Chromogenic Solution. And incubating in the dark for 10 min at 37 °C. Centrifuge at 12,000 ×$g$ for 10 min at 4 °C to remove insoluble material. Approximately 300 μL reacted solution were added to a 96-well plate for iron measurement. Finally, the absorbance was measured at 593 nm, and a standard curve line was used for iron concentration calculation.

## Histological analysis
Histopathological analysis was performed as previously described methods[56]. Liver tissues were fixed in 10% neutral buffered formalin at room temperature, embedded in paraffin and cut into 3.5 μm sections, and then stained with hematoxylin and eosin (H&E). Images were captured by using a light microscope (Olympus, Tokyo, Japan) to analyze liver damage. Descriptions of pathological changes of liver tissues included, but were not limited to, necrosis, inflammation, ballooning, etc. Typical lesion areas in the image are marked with arrows or dashed circles in the report. Under light microscopy, necrosis manifests as detectable hepatocyte pyknosis, nuclear fragmentation, and karyolysis, which is indicated by yellow arrows or dashed circles in the images. Neutrophil infiltration can be seen under light microscopy, and the cell nuclei are dark purple, which is indicated by

green arrows or dashed circles in the image. Under light microscopy, ballooning degeneration manifests as hepatocytes swelling into spherical shapes with almost transparent cytoplasm, which is indicated by black arrows in the image.

## Isolation and culture of primary hepatocytes
Primary hepatocytes were obtained from 6- to 8-week-old mice by using a modified two-step collagenase perfusion method. Briefly, mice were anesthetized and perfused with Hank's balanced salt solution (HBSS, without $Ca^{2+}$ and $Mg^{2+}$) (abs9257, Absin, Shanghai China) containing 0.5 mM EGTA (E8050, Solarbio, Beijing, China) and digested with HBSS containing 0.04% collagenase Type IV (17104019, Gibco, New York, USA) via the portal vein. Then, the livers were excised, minced, and filtered. The cells were centrifuged 3 times at $50 \times g$ for 4 min each to purify hepatocytes. Isolated hepatocytes were then cultured in Dulbecco's Modified Eagle Medium Nutrient Mixture F-12 (DMEM/F12) (C11330500BT, Gibco, Grand Island, NY) supplemented with 10% fetal bovine serum (10099141C, Gibco, Grand Island, NY) and 1% penicillin-streptomycin (15140-122, Gibco, Grand Island, NY) at 37 °C with an environment of 5% $CO_2$.

## Cell culture and treatment
Alpha mouse liver 12 (*AML12*) cells were purchased from Procell Life (CL-0602, Wuhan, China). The *AML12* cells were cultured in DMEM/F12 (with 10% FBS, 1% penicillin/streptomycin, 10 μg/mL insulin, 5.5 μg/mL transferrin, 5 ng/mL selenium, and 40 ng/mL dexamethasone) in a humidified environment at 37 °C and 5% $CO_2$. Unconjugated bilirubin (UCB) (14370, Sigma-Aldrich, St. Louis, MO, USA) was utilized in the treatment of cells, and was prepared according to the article by Granato et al.[57]. The UCB solution was diluted with DMEM/F12 medium containing 5% FBS to prepare working solutions of 50 μM, 100 μM, and 200 μM. The molar ratios of UCB to albumin were 2.75, 5.5, and 11, respectively. In this study, OA, BMS-986020 (BMS) (HY-100619, Med Chem Express, NJ, USA), MK-571 and resveratrol (Res) (R5010, Sigma-Aldrich, St. Louis, MO, USA) were added 2 h before UCB (100 μM) administration.

## siRNA transfection
The Nrf2 siRNA kit was purchased from Ruibo Biotechnology Co., Ltd (Guangzhou, China). *AML12* cells were transfected with siRNA according to the manufacturer's protocol. *AML12* cells were seeded at a density of $0.6 \times 10^5$ cells/well in 12-well plates, and cells were transfected with Nrf2-siRNA using Lipofectamine 2000 reagent at a final concentration of 50 nmol/L. Then, cells were incubated with normal medium for 48 h to determine the knockdown efficiency. The specific synthesized Nrf2 siRNA sequence is: GATGGACTTGGAGTTGCCA.

## Lentiviral transfection
HBLV-ZsGreen-PURO fragment was selected as the cloning vector. The titer of lentivirus was maintained at $4 \times 10^8$. Lentivirus (LV) containing SOD2 was purchased from Hanbio (Shanghai, China). HBLV-Scramble or HBLV-SOD2 (MOI: 50) was transfected into *AML12* cells according to the manufacturer's protocol. Then, cells were selected with puromycin (6 μg/mL). Green fluorescence and western blotting were conducted to determine the transfection efficiency of LV-SOD2 in *AML12* cells.

## Real-time quantitative polymerase chain reaction analysis
Total RNA was prepared from cultured cells, mouse liver tissues and human blood samples using TRIzol® reagent (Takara, Tokyo, Japan) according to the manufacturer's instructions. Total RNA content and purity were measured by Ultra Micro Spectrophotometer (ND2000, Thermo Fisher Scientific, MA, USA). cDNA was synthesized with 1 μg RNA using the PrimeScript™ RT Reagent Kit (Takara, Tokyo, Japan). The real-time qPCR apparatus (CFX Connect, Bio-Rad, CA, USA) was used for PCR amplification. The *GAPDH* was used as the internal reference gene for data normalization. Specific primer sequences for all genes are shown in Supplementary Table 2.

## Western blotting analysis

Liver tissues (30-50 mg) and cultured cells were prepared in lysis buffer containing 1% of PMSF and 1% phosphatase inhibitor, followed by centrifugation ($12,000 \times g$, 10 min, 4 °C), and the protein concentration was determined by the BCA Protein Assay Kit (P0012, Beyotime, Shanghai, China). Thirty micrograms of protein per lane were separated by 10% SDS-PAGE gels and electrophoretically transferred onto polyvinylidene fluoride (PVDF) membranes. Membranes are further blocked with 5% skimmed milk for 2 h at room temperature and incubated overnight with primary antibodies at 4 °C. Then, the membranes were incubated with appropriate horseradish peroxidase (1:5000, Proteintech, China) conjugated secondary antibodies for 50 min at room temperature. The protein blots were detected by enhanced chemiluminescence (ECL) reagent (7 Sea Biotech, China) with a chemiluminescence detection system (Bio-Rad, Chemi Doc, CA, USA). The antibodies used are listed in Supplementary Table 3.

## Immunohistochemistry

Mouse liver tissues were fixed in 4% formalin overnight and embedded in paraffin. Tissue slicing (3 μm) dewaxing and antigen (citrate) repair were performed. Tissue sections were treated with 3% hydrogen peroxide (PV-6001, ZSGB-BIO, Beijing, China) for 15 min, and blocked with goat serum for 30 min. Tissue sections were incubated with the primary antibody anti-HO-1 (1:200) overnight at 4 °C. After overnight incubation, the slides were washed and the sections were incubated with appropriate horseradish peroxidase (1:5000, Proteintech, China) conjugated secondary antibody for 30 min at room temperature. Then the slices were incubated with diaminobenzidine (DAB) (ZLI-9018, ZSGB-BIO, Beijing, China) for 3 min, stained with hematoxylin for 3 min, rinsed for 10 min, and then dehydrated and sealed with neutral gum. Finally, the sections were visualized by a microscope at a magnification of ×40.

## Mitochondrial ROS analysis

MitoSOX (M36008, Molecular Probes, SV, USA) is commonly used in cellular systems to detect mitochondrial ROS formation[58]. Upon the completion of treatment, cells were incubated with 8 μM Mito-SOX Red probe at 37 °C for 45 min, and then washed three times with PBS. Images were captured by using an inverted fluorescence microscope (Olympus, Tokyo, Japan) and analyzed with the ImageJ software (National Institutes of Health).

## Analysis of mitochondrial membrane potential (ΔΨm)

After the completion of the treatment, cells were washed with PBS and stained with JC-1 (M8650, Solarbio, Beijing, China) according to the manufacturer's instructions[59]. The fluorescence was detected using a spectral scanning multifunctional microplate reader (Vorioskan Lux, Thermo Fisher Scientific, MA, USA). Data was analyzed by measuring the ratio of red to green fluorescence.

## Measurement of oxygen consumption

Oxygen consumption of *AML12* cells was measured at 37 °C in a high-resolution oxygraph (Oxygraph-2k Oroboros Instruments, Austria)[60]. Briefly, cells ($8.0 \times 10^5$ cells) were seeded in 60 mm culture plates and treated with UCB (50 and 100 μM) for 12 h. Upon the completion of the treatments, cells were harvested via standard trypsinization, and were then resuspended in DMEM/F12 medium without serum for respirometry. For the measurements of oxygen consumption, 2.5 mL of cell suspension ($2 \times 10^6$ cells) was applied to the closed chamber with a magnetic stirring. The oxygen consumption was continuously recorded for 30 min. Oxygen flux (pmol $O_2$/s/$10^6$ cells), being directly proportional to oxygen consumption, was recorded and calculated using DatLab software.

## Statistics and reproducibility

All graphs display individual data points, representing individual readings for each sample. The legend specifies the sample size and number of repetitions. All data are shown as mean ± standard error (SEM). Before analyzing all results, the Shapiro-Wilk test was conducted to determine normality. If the Shapiro-Wilk test result was $p > 0.05$, the data was deemed to be close to normal distribution. Student's t-test was used for statistical evaluation of the comparison between two groups that had a normal distribution. Data from multiple groups were compared using one-way ANOVA if they followed a normal distribution. Tukey's post hoc test was used for analysis of data meeting the homogeneity of variance assessment, or Games Howell analysis for heterogeneous data was used for data from multiple groups if they followed a normal distribution. If the Shapiro-Wilk test result was $p < 0.05$, the distribution was not normal, and nonparametric tests were employed. Image Lab was used to process and quantitatively evaluate protein bands. The specific RNA-seq analysis method can be found in the supplemental materials. SPSS (version 23.0, SPSS Inc., USA) and GraphPad prism (version 8.0, GraphPad Inc., USA) were used for statistical analyses. Differences were considered statistically significant at $p < 0.05$.

## Reporting summary

Further information on research design is available in the Nature Portfolio Reporting Summary linked to this article.

## Data availability

The RNA sequencing data has been deposited in GEO under the accession number GSE263093. All other data supporting the results and conclusions of this paper are available in the paper and its Supplementary Information. The source data behind the graphs can be found in Supplementary Data 1. The processed sequencing data can be found in Supplementary Data 2. The source data of human samples (including the source data for Fig. 7, Supplementary Table 1 and samples information) can be found in Supplementary Data 3. Uncropped images of the gel can be found in Supplementary Fig. 9.

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

## Acknowledgements

We thank Dr. Lili Ji from Shanghai University of Traditional Chinese Medicine for providing *Nrf2*$^{-/-}$ mice, and Wanlan Shi and Yingying Li from the Key Laboratory of Basic Pharmacology of Ministry of Education of Zunyi Medical University for their care of animals and their assistance in tissue processing. The work was supported by the National Natural Science Foundation of China (Grant No. 81760678, and 82260806), First-Class Disciplines Fund of the Education Department of Guizhou Province (No. GNYL [2017] 006 YLXKJS-YS-05) and Zunyi Science and Technology Bureau Project (No. [2021]-192).

## Author contributions

S.Z. and Y.L. conceived the overall project; Y.W., X.F., Y.L. and S.Z. designed the studies, analyzed data, and wrote the manuscript; Y.W., X.F., L.Z., Y.H., R.G., S.X., S.L., J.H. and Y.Y. conducted experiments and provided technical support. S.Z. and J.K. critically revised the manuscript. H.J. and J.L. collected clinical information and samples. S.Z. and Y.L. were responsible for obtaining funding. All authors reviewed and approved the manuscript.

## Competing interests

The authors declare no competing interests.
