## [Peer Review File · Communications Biology]

Reviewers' comments:

Reviewer #1 (Remarks to the Author):

In this study, Wang et al. show that the Nrf2 antioxidant response is not always hepatoprotective, as it exacerbates liver injury in chemical-induced cholestasis caused by ANIT and OA. In this context, the authors focus on the importance of HO-1, the rate-limiting enzyme of heme catabolism to biliverdin/bilirubin, which is induced upon Nrf2 activation. The authors claim that in cholestasis, oxidative stress activates Nrf2/HO-1, leading to bilirubin accumulation and mitochondrial impairment in hepatocytes. High levels of bilirubin reciprocally upregulate Nrf2 and HO-1 activation, forming a vicious cycle that promotes liver injury.

These findings are interesting, as they support a novel pathogenic mechanism in the context of chemical-induced cholestasis. However, there are important questions and issues that must be resolved and prevent the acceptance of this manuscript in its present form:

- An issue of major concern is that the authors concluded that activation of hepatic Nrf2/HO-1 results in excessive accumulation of bilirubin in the liver. Approximately 80% of bilirubin is made from the breakdown of hemoglobin from senescent red blood cells, which primarily occurs in the cells of the reticuloendothelial system, where HO-1 is present in high concentrations. Within the liver, HO-1 is highly expressed in Kupffer cells, but not so high in hepatocytes. How do authors conciliate their hypothesis with the fact that most of the bilirubin (UCB) is not produced in hepatocytes? In this study, bilirubin levels were measured in the serum but not in the liver. If bilirubin synthesis increases in the liver, where does the substrate heme come from? The authors should demonstrate that the increase in liver/hepatocyte HO-1 significantly contributes to higher liver/hepatocyte bilirubin levels.

- Regarding bile acids: Why is the TBA level decreased in Fig. 1e-ANIT when HO-1 is inhibited by ZnPP? HO-1 is not involved in BA metabolism. Moreover, the results of the ANIT and OA models are different. Finally, in Fig. S1e (CCl₄ model), ZnPP triggered an increase in TBA, which seems contradictory to the ANIT model.

- If we compare Fig 1e with Fig.2j, it can be concluded that, in the OA model, the level of TBA is not dependent on HO-1, but dependent on Nrf2. If so, what is the Nrf2-dependent mechanism that leads to increased TBA levels? Something similar can be said about TBIL and DBIL, knocking out Nrf2 was much more effective than inhibiting HO-1.

- The authors argue that bilirubin is a major contributor to liver injury but ignore the potential contribution of bile acids. Thus, the contribution of bile acids should also be tested in a manner similar to bilirubin in the experiments shown in Fig. 5.

- Fig 3 h and i: Please also provide the DBIL/TBIL ratio. Despite better excretion, an increase in this ratio is expected when more UDPGA is provided.

- Regulation of transporters: As the authors state, hepatobiliary transport dysfunction is considered the major cause of cholestasis, and hence of CLI. The authors demonstrated that the major bile acid and bilirubin transporters were downregulated in the OA model both in vivo and in vitro (Fig. 4a, Fig. S3). This could be, in fact, more important than the bilirubin futile cycle presented in Fig 7k, as it seems

feasible that if the transporters were not repressed, bilirubin and bile acids would not accumulate. The authors should provide evidence of the potential role of the Nrf2/HO-1 pathway in transporter repression (see for example, 10.1016/j.fct.2021.112664). It is also important to measure the protein expression levels of transporters in Nrf2 KO mice, and add the results to Fig. 2m.

- In the same context, it is important to note that, according to previous studies, activation of Nrf2 leads to Mrp2 upregulation, via an ARE element located at the Mrp2 proximal promoter region (direct target gene) (10.1042/BJ20051518). However, in the present study, Mrp2 was repressed upon OA-induced Nrf2 activation.

- Similarly, please discuss previous results that conflict with your conclusions, for example, 10.1016/j.bbrc.2009.08.156.

- I do not understand the rationale for using a BSEP inhibitor (Fig. 5b). BSEP is not involved in bilirubin transport, and consequently, I do not understand how this inhibitor affects the toxicity of bilirubin. Moreover, BMS-986020 has been reported to also inhibit mitochondrial function (basal and maximal respiration, ATP production, and spare capacity) in human hepatocytes, suggesting a number of mixed effects in the results shown in Fig 5b.

- Finally, additional patient information is required. What cholestatic diseases were considered? A table with patient information, including the pathological diagnosis, is required. Is the pathology the same in all patients with CLI or within each severity group? Moreover, a high bilirubin level is not always a valid indicator of cholestasis, as it is also elevated in severe non-cholestatic liver diseases. Considering both bilirubin and ALP or GGT levels is likely a better approach for patient stratification. It is also important to note that table s4 has not been mentioned in the text.

Reviewer #2 (Remarks to the Author):

In the present investigation, Wang et al. investigated the role of Nrf2/HO-1 in cholestatic liver injury (CLI). They found that Nrf2/HO-1 activation enhances liver injury rather than protects from CLI with excessive bilirubin accumulation. Inhibiting HO-1 or ameliorating bilirubin transport alleviates liver injury in CLI models. In addition, Nrf2 knockout resulted in hepatoprotection in CLI mice. In contrast, Nrf2 knockout aggravated liver damage in non-CLI mice. In CLI, Nrf2/HO-1 induces bilirubin accumulation and impairs mitochondrial function. Furthermore, the expression of Nrf2 and HO-1 was elevated in the serum of patients with CLI, and the HO-1 expression was correlated with the severity of CLI and bilirubin levels.

The study by Wang et al. is interesting, well-written, and provides intriguing insights into the role of the Nrf2/HO-1 axis in the liver. The experiments were properly and logically conducted. The data are solid and support the conclusions drawn.

Minor issue:

- The histology pictures are minuscule and hard to evaluate. New, more detailed, and bigger pictures should be provided at higher magnification.

Reviewer #3 (Remarks to the Author):

COMMSBIO-23-3439-T

Activation of Nrf2/HO-1 signaling pathway exacerbates cholestatic liver injury

This is a research article that tries to explain the toxic functions of Nrf2/HO-1 signaling pathway in the hepatocyte cholestatic injuries. The manuscript consists of many experiments separately performed in both mice and cultured cells. It sounds energetic but includes some fundamental issues that undermine reliability of the study. Further, the text is too redundant to go through and understand.

1. The results of mice experiments, consisting most parts of the study and generating an initial question, are not very convincing because all the photomicrographs are too low qualities to evaluate the toxic or protective effects of OA and others. No pathological criteria of hepatocyte edema and large-scale necrosis is defined clearly. What kind of inflammatory cells are infiltrated in the liver? In such a study of experimental toxicology, histological evaluation would be very essential, and therefore, should be very clearly presented as first materials for the progression of experiments.
2. What is the meaning to measure the expression of Nrf2, a transcriptional factor, in the patient serum? Where does the Nrf2 expression come from?
3. The results section is too long and very annoying to understand the experimental strategy and significance of the results. The results should be concisely described.
4. Most of the results obtained are phenomenological but mechanistic.
5. Some parts of the discussion are merely repeated description of the results. It does not include a discussion or show only too speculative comments.

Reviewers' comments:

Reviewer #1 (Remarks to the Author):

In this study, Wang et al. show that the Nrf2 antioxidant response is not always hepatoprotective, as it exacerbates liver injury in chemical-induced cholestasis caused by ANIT and OA. In this context, the authors focus on the importance of HO-1, the rate-limiting enzyme of heme catabolism to biliverdin/bilirubin, which is induced upon Nrf2 activation. The authors claim that in cholestasis, oxidative stress activates Nrf2/HO-1, leading to bilirubin accumulation and mitochondrial impairment in hepatocytes. High levels of bilirubin reciprocally upregulate Nrf2 and HO-1 activation, forming a vicious cycle that promotes liver injury.

These findings are interesting, as they support a novel pathogenic mechanism in the context of chemical-induced cholestasis. However, there are important questions and issues that must be resolved and prevent the acceptance of this manuscript in its present form:

- An issue of major concern is that the authors concluded that activation of hepatic Nrf2/HO-1 results in excessive accumulation of bilirubin in the liver. Approximately 80% of bilirubin is made from the breakdown of hemoglobin from senescent red blood cells, which primarily occurs in the cells of the reticuloendothelial system, where HO-1 is present in high concentrations. Within the liver, HO-1 is highly expressed in Kupffer cells, but not so high in hepatocytes. How do authors conciliate their hypothesis with the fact that most of the bilirubin (UCB) is not produced in hepatocytes? In this study, bilirubin levels were measured in the serum but not in the liver. If bilirubin synthesis increases in the liver, where does the substrate heme come from? The authors should demonstrate that the increase in liver/hepatocyte HO-1 significantly contributes to higher liver/hepatocyte bilirubin levels.

- Regarding bile acids: Why is the TBA level decreased in Fig. 1e-ANIT when HO-1 is inhibited by ZnPP? HO-1 is not involved in BA metabolism. Moreover, the results of the ANIT and OA models are different. Finally, in Fig. S1e (CCl4 model), ZnPP triggered an increase in TBA, which seems contradictory to the ANIT model.

- If we compare Fig 1e with Fig.2j, it can be concluded that, in the OA model, the level of TBA is not dependent on HO-1, but dependent on Nrf2. If so, what is the Nrf2-dependent mechanism that leads to increased TBA levels? Something similar can be said about TBIL and DBIL, knocking out Nrf2 was much more effective than inhibiting HO-1.

- The authors argue that bilirubin is a major contributor to liver injury but ignore the potential contribution of bile acids. Thus, the contribution of bile acids should also be tested in a manner similar to bilirubin in the experiments shown in Fig. 5.

- Fig 3 h and i: Please also provide the DBIL/TBIL ratio. Despite better excretion, an increase in this ratio is expected when more UDPGA is provided.

- Regulation of transporters: As the authors state, hepatobiliary transport dysfunction is considered the major cause of cholestasis, and hence of CLI. The authors demonstrated that the major bile acid and bilirubin transporters were downregulated in the OA model both in vivo and in vitro (Fig. 4a, Fig. S3). This could be, in fact, more important than the bilirubin futile cycle presented in Fig 7k, as it seems feasible that if the transporters were not repressed, bilirubin and bile acids would not accumulate. The authors should provide evidence of the potential role of the Nrf2/HO-1 pathway in transporter repression (see for example, 10.1016/j.fct.2021.112664). It is also important to measure the protein expression levels of transporters in Nrf2 KO mice, and add the results to Fig. 2m.

- In the same context, it is important to note that, according to previous studies, activation of Nrf2 leads to Mrp2 upregulation, via an ARE element located at the Mrp2 proximal promoter region (direct target gene) (10.1042/BJ20051518). However, in the present study, Mrp2 was repressed upon OA-induced Nrf2 activation.

- Similarly, please discuss previous results that conflict with your conclusions, for example, 10.1016/j.bbrc.2009.08.156.

- I do not understand the rationale for using a BSEP inhibitor (Fig. 5b). BSEP is not involved in bilirubin transport, and consequently, I do not understand how this inhibitor

affects the toxicity of bilirubin. Moreover, BMS-986020 has been reported to also inhibit mitochondrial function (basal and maximal respiration, ATP production, and spare capacity) in human hepatocytes, suggesting a number of mixed effects in the results shown in Fig 5b.

- Finally, additional patient information is required. What cholestatic diseases were considered? A table with patient information, including the pathological diagnosis, is required. Is the pathology the same in all patients with CLI or within each severity group? Moreover, a high bilirubin level is not always a valid indicator of cholestasis, as it is also elevated in severe non-cholestatic liver diseases. Considering both bilirubin and ALP or GGT levels is likely a better approach for patient stratification. It is also important to note that table s4 has not been mentioned in the text.

Reviewer #2 (Remarks to the Author):

In the present investigation, Wang et al. investigated the role of Nrf2/HO-1 in cholestatic liver injury (CLI). They found that Nrf2/HO-1 activation enhances liver injury rather than protects from CLI with excessive bilirubin accumulation. Inhibiting HO-1 or ameliorating bilirubin transport alleviates liver injury in CLI models. In addition, Nrf2 knockout resulted in hepatoprotection in CLI mice. In contrast, Nrf2 knockout aggravated liver damage in non-CLI mice. In CLI, Nrf2/HO-1 induces bilirubin accumulation and impairs mitochondrial function. Furthermore, the expression of Nrf2 and HO-1 was elevated in the serum of patients with CLI, and the HO-1 expression was correlated with the severity of CLI and bilirubin levels.

The study by Wang et al. is interesting, well-written, and provides intriguing insights into the role of the Nrf2/HO-1 Nrf2/HO-1 axis in the liver. The experiments were properly and logically conducted. The data are solid and support the conclusions drawn.

Minor issue:

- The histology pictures are minuscule and hard to evaluate. New, more detailed, and bigger pictures should be provided at higher magnification.

Reviewer #3 (Remarks to the Author):

COMMSBIO-23-3439-T

Activation of Nrf2/HO-1 signaling pathway exacerbates cholestatic liver injury

This is a research article that tries to explain the toxic functions of Nrf2/HO-1 signaling pathway in the hepatocyte cholestatic injuries. The manuscript consists of many experiments separately performed in both mice and cultured cells. It sounds energetic but includes some fundamental issues that undermine reliability of the study. Further, the text is too redundant to go through and understand.

1. The results of mice experiments, consisting most parts of the study and generating an initial question, are not very convincing because all the photomicrographs are too low qualities to evaluate the toxic or protective effects of OA and others. No pathological criteria of hepatocyte edema and large-scale necrosis is defined clearly. What kind of inflammatory cells are infiltrated in the liver? In such a study of experimental toxicology, histological evaluation would be very essential, and therefore, should be very clearly presented as first materials for the progression of experiments.
2. What is the meaning to measure the expression of Nrf2, a transcriptional factor, in the patient serum? Where does the Nrf2 expression come from?
3. The results section is too long and very annoying to understand the experimental strategy and significance of the results. The results should be concisely described.
4. Most of the results obtained are phenomenological but mechanistic.
5. Some parts of the discussion are merely repeated description of the results. It does not include a discussion or show only too speculative comments.

We would like to express our sincere appreciation to the editor and the reviewers for the valuable and constructive and insightful comments concerning our article entitled "activation of Nrf2/HO-1 signaling pathway exacerbates cholestatic liver injury". These comments are all valuable and helpful for improving our article. According to the editor and reviewers' comments, we have made extensive modifications to our manuscript and supplemented extra data to make our results convincing (please refer to one-by-one responses below). In this revised version, changes to our manuscript were all highlighted within the document by using red-colored text. Additional data supplemented include: 1) We detected the liver bilirubin levels of mice in ANIT and ZnPP+ANIT groups using liquid chromatography-mass spectrometry (LC-MS); 2) We detected the expression of HO-1 in mouse liver by immunohistochemistry; and 3) We provide new, more detailed and larger histological images.

The following are our one-by-one responses to the reviewers' comments.

Reviewer #1 (Remarks to the Author):

In this study, Wang et al. show that the Nrf2 antioxidant response is not always hepatoprotective, as it exacerbates liver injury in chemical-induced cholestasis caused by ANIT and OA. In this context, the authors focus on the importance of HO-1, the rate-limiting enzyme of heme catabolism to biliverdin/bilirubin, which is induced upon Nrf2 activation. The authors claim that in cholestasis, oxidative stress activates Nrf2/HO-1, leading to bilirubin accumulation and mitochondrial impairment in hepatocytes. High levels of bilirubin reciprocally upregulate Nrf2 and HO-1 activation, forming a vicious cycle that promotes liver injury.

These findings are interesting, as they support a novel pathogenic mechanism in the context of chemical-induced cholestasis. However, there are important questions and issues that must be resolved and prevent the acceptance of this manuscript in its present form:

- An issue of major concern is that the authors concluded that activation of hepatic Nrf2/HO-1 results in excessive accumulation of bilirubin in the liver. Approximately 80% of bilirubin is made from the breakdown of hemoglobin from senescent red blood cells, which primarily occurs in the cells of the reticuloendothelial system, where HO-1 is

present in high concentrations. Within the liver, HO-1 is highly expressed in Kupffer cells, but not so high in hepatocytes. How do authors conciliate their hypothesis with the fact that most of the bilirubin (UCB) is not produced in hepatocytes? In this study, bilirubin levels were measured in the serum but not in the liver. If bilirubin synthesis increases in the liver, where does the substrate heme come from? The authors should demonstrate that the increase in liver/hepatocyte HO-1 significantly contributes to higher liver/hepatocyte bilirubin levels.

Response: Thank you for taking the time to review this manuscript. We thank you for your comments and suggestions, which have helped clarify and improve our manuscript. Specific responses to the comments raised above are as follow:

(1) We agree with the opinion of the reviewing expert that approximately 80% of bilirubin is made from the breakdown of hemoglobin from senescent red blood cells, which primarily occurs in the cells of the reticuloendothelial system, where HO-1 is present in high concentrations. Bilirubin is usually produced in the reticuloendothelial system of the liver, spleen, bone marrow, and other organs. The liver has a very rich blood supply. Under certain conditions, erythrocytes in blood are broken down in the liver by reticuloendothelial cells causing the release of globin and heme. The heme ring is opened by heme oxygenase (HMOX) of the liver, releasing iron (Fe) and carbon monoxide (CO), and the ring structure is converted to biliverdin. Subsequently, biliverdin reductase (BVR) reduces biliverdin to bilirubin. And the bilirubin produced by other organs binds to albumin in the blood, and flows with the blood to the liver. Subsequently, bilirubin is transported into the hepatocyte by the organic anion transporters, OATP1 and OATP2. UDP-glucuronosyltransferase (UGT1A1) adds glucuronic acid to the unconjugated bilirubin forming a soluble bilirubin that permits secretion. The above can lead to two conclusions: 1) The liver produces bilirubin, and 2) The liver takes up bilirubin. But when the liver bilirubin content increases to the point where UGT1A1 cannot be metabolized or efflux is blocked, it can lead to excessive accumulation of bilirubin in the liver, ultimately leading to liver damage. Our conclusion does not conflict with the fact that most bilirubin (UCB) is not produced in hepatocytes.

(2) The liver has a very rich blood supply. Under certain conditions, erythrocytes in blood are broken down in the liver by reticuloendothelial cells causing the release of globin and heme.

(3) We used immunohistochemistry and LC-MS to detect the expression of HO-1 in

mouse liver and the level of bilirubin in mouse liver, respectively, to make our results convincing and present them in Figures 1h and 1k in the revised manuscript. LC-MS and immunohistochemistry methods are introduced in lines 519-534, and 610-620, respectively. The results are described in lines 132-138, and marked in Figure 1k and explained in the legend (lines 845-850). From the results of immunohistochemistry analysis, OA induces an increase in the expression of HO-1, and it can be seen that HO-1 is not only expressed in Kupffer cells, but also in hepatocytes (Fig. 1k). Simultaneously, the increase in liver HO-1 is significantly correlated to higher liver bilirubin levels (Fig. 1h). The results are as follows:

Fig. 1 Increased HO-1 expression is involved in pathological changes in CLI. a

Representative H&E staining of liver tissues in mice (6- to 8-week-old) after administration of ZnPP, followed by 295.9 mg/kg OA or 60 mg/kg ANIT (n = 5). **Neutrophil infiltration (green arrows) and hepatocyte necrosis (yellow arrow or dashed circle)**. Scale bar, 100 μ m or 50 μ m. b-g Serum levels of ALT, AST, ALP, TBA, TBIL, and DBIL (n = 5). h **Liver UCB concentration (n=5)**. i, j Expression of HO-1 protein and mRNA after oral administration of OA. k **Immunohistochemical staining of liver HO-1. Scale bar, 50 μ m. Kupffer cells (green triangle) and hepatocytes (yellow triangle)**. l Expression of HO-1 protein after intragastric administration of ANIT.

- Regarding bile acids: Why is the TBA level decreased in Fig. 1e-ANIT when HO-1 is inhibited by ZnPP? HO-1 is not involved in BA metabolism. Moreover, the results of the ANIT and OA models are different. Finally, in Fig. S1e (CCl₄ model), ZnPP triggered an increase in TBA, which seems contradictory to the ANIT model.

Response: Thanks for the reviewer's comments. Specific responses to the comments raised by the reviewer are as follows:

(1) Studies have shown that bilirubin bile pigments may affect the synthesis or secretion of cholesterol, thereby affecting the synthesis of bile acids (10.1152/ajpendo.00396.2016, 10.1038/srep09886). The different mechanism of action or degree of injury between OA and ANIT may lead to different outcomes of TBA in ANIT and OA models.

(2) Comparing Fig. 1e and Fig. S1e, it can be seen that the increase in TBA caused by ZnPP in the CCl₄ model is slight. In the cases of severe impairment of hepatocytes or hepatocyte functions, the liver may reduce the uptake or reuptake of bile acids, resulting in slightly increased serum TBA levels.

(3) Meanwhile, we have revised the corresponding text in the manuscript based on the suggestions put forward by the reviewer. We added the following sentences in Results. L119-123: 'The above results indicate that HO-1 inhibitors can improve OA induced liver injury, which is consistent with previous reports^{19,20}. Notably, HO-1 is the rate limiting enzyme for bilirubin synthesis, and the inhibition of HO-1 resulted in a significantly decrease in bilirubin synthesis.'

L139-142: 'With one exception, the TBA level decreased after inhibition of HO-1 (Fig. 1e), which was inconsistent with the results of the OA model. This may be due to the different degrees of liver damage or the regulatory effect between bilirubin and bile

acids.'

- If we compare Fig 1e with Fig.2j, it can be concluded that, in the OA model, the level of TBA is not dependent on HO-1, but dependent on Nrf2. If so, what is the Nrf2-dependent mechanism that leads to increased TBA levels? Something similar can be said about TBIL and DBIL, knocking out Nrf2 was much more effective than inhibiting HO-1.

Response: Thanks for this reviewer's comments. From Figure 2b and m, it can be seen that the increase in HO-1 protein expression caused by OA is mediated by Nrf2. Nrf2 knockout blocked the effect of OA on HO-1, while HO-1 inhibitors only partially inhibited the activity of HO-1 and did not completely block its effect. Therefore, it is reasonable that knocking out Nrf2 is more effective than inhibiting HO-1. We have revised the corresponding text to reflect this observation. Please refer to L174-177: 'In addition, Nrf2 knockout significantly blocked the regulatory effect of OA on HO-1, and was more effective than inhibiting HO-1 in reducing bilirubin accumulation and improving liver damage.'

Research has shown that 1) bilirubin may affect the synthesis or secretion of cholesterol, thereby affecting the synthesis of bile acids (10.1152/ajpendo.00396.2016, 10.1038/srep09886), and 2) δ -ALAS1 mRNA expression increased significantly upon activation of the farnesoid X receptor (FXR) with chenodeoxycholic acid (CDCA), suggesting that bile acids positively regulate heme synthesis genes (10.1002/hep.21879). All of the above indicate that there may be mutual regulation between bile acids and bilirubin. However, more research is needed on this issue.

- The authors argue that bilirubin is a major contributor to liver injury but ignore the potential contribution of bile acids. Thus, the contribution of bile acids should also be tested in a manner similar to bilirubin in the experiments shown in Fig. 5.

Response: Thank you for the reviewer's comments. Throughout the process, we do not deny the role played by bile acids. Our research group has also studied the contribution of bile acids to liver damage and published relevant results (10.1002/jat.4298, 10.1002/jat.4456). However, in this study, we focus on the role of bilirubin.

- Fig 3 h and i: Please also provide the DBIL/TBIL ratio. Despite better excretion, an increase in this ratio is expected when more UDPGA is provided.

Response: We appreciate the valuable feedback and questions raised by this reviewer. In this revised manuscript, we provide the IBIL in Figure 3j. The main toxic agent is indirect bilirubin (IBIL) (also known as unconjugated bilirubin), so calculating IBIL (TBIL minus DBIL) may be more intuitive. We have revised the corresponding text accordingly. Please refer to lines 202: ‘indirect bilirubin (IBIL).’

- Regulation of transporters: As the authors state, hepatobiliary transport dysfunction is considered the major cause of cholestasis, and hence of CLI. The authors demonstrated that the major bile acid and bilirubin transporters were downregulated in the OA model both in vivo and in vitro (Fig. 4a, Fig. S3). This could be, in fact, more important than the bilirubin futile cycle presented in Fig 7k, as it seems feasible that if the transporters were not repressed, bilirubin and bile acids would not accumulate. The authors should provide evidence of the potential role of the Nrf2/HO-1 pathway in transporter repression (see for example, 10.1016/j.fct.2021.112664). It is also important to measure the protein expression levels of transporters in Nrf2 KO mice, and add the results to Fig. 2m.

Response: Thanks for this reviewer’s comments. We agree that transporters play a significant role in this process of cholestasis. Using the chemical induced cholestasis models (modeled with both OA and ANIT), our studies in the present project focused on the Nrf2/HO-1/bilirubin axis. Particularly, we found that activation of the Nrf2/HO-1 signaling pathway exacerbates liver injuries in the cholestasis models, and one of the main tasks in the present study was to dissect the role of bilirubin in the worsening effect and its mechanism. We agree with the reviewer that transporters also play an important role that may be entangled in the mechanism, and we hope to elucidate more detailed mechanism in the near future.

- In the same context, it is important to note that, according to previous studies, activation of Nrf2 leads to Mrp2 upregulation, via an ARE element located at the Mrp2 proximal promoter region (direct target gene) (10.1042/BJ20051518). However, in the

present study, Mrp2 was repressed upon OA-induced Nrf2 activation.

Response: Thanks for this reviewer's comments. We agree that the activation of Nrf2 may lead to an upregulation of Mrp2. However, Mrp2 is also under regulation of other factors, for example, farnesoid X receptor (FXR), a nuclear receptor. We have previously reported that OA treatment inhibits FXR expression, which may directly cause downregulation of Mrp2 (doi.org/10.1002/jat.4298). In addition, the phosphorylation of Ezrin Thr567 mentioned in 10.1016/j.ghep.2015.07.016 or the Slug mentioned in 10.3390/biom12060806 can also regulate the expression of Mrp2. Therefore, the activation of Nrf2 may exert different effect on Mrp2 under different circumstances, and in our study, the Nrf2 activation induced by OA results in the repression of Mrp2.

- Similarly, please discuss previous results that conflict with your conclusions, for example, 10.1016/j.bbrc.2009.08.156.

Response: Thanks for this reviewer's comments. In view of the fact that our conclusion conflicts with previous results, we have the following aspects as arguments for our results, which make our results more convincing. 1) Both OA and ANIT are chemical drugs, and our model (with OA and ANIT) is chemically induced cholestatic liver injury, which differs from the mechanism of action of bile duct ligation as reported in the studies showing hepatoprotective role of sustained Nrf2 activation against liver injury associated with cholestasis (10.1016/j.bbrc.2009.08.156). 2) There have been reports showing that the overexpression of HO-1 accelerates the progression of chronic cholestasis to liver fibrosis (10.3748/wjg.v19.i19.2921, 10.3748/wjg.v13.i25.3478). 3) Knocking out Nrf2 in mice can prevent bile duct ligation-induced cholestasis from causing more serious liver injury (10.1152/ajpgi.00263.2011). All these are supportive of our conclusions. In this revised manuscript, we have extended discussion in this matter and revised the corresponding text in the Discussion section.

Please refer to L395-401: 'This seems to contradict the previous results showing that sustained Nrf2 activation has a hepatoprotective effect on cholestasis-related liver injur. However, our model (OA or ANIT) is chemically induced cholestatic liver injury, which differs from the mechanism of action of bile duct ligation. In addition, there are studies demonstrating that the overexpression of HO-1 accelerates the progression of chronic cholestasis to liver fibrosis.'

- I do not understand the rationale for using a BSEP inhibitor (Fig. 5b). BSEP is not involved in bilirubin transport, and consequently, I do not understand how this inhibitor affects the toxicity of bilirubin. Moreover, BMS-986020 has been reported to also inhibit mitochondrial function (basal and maximal respiration, ATP production, and spare capacity) in human hepatocytes, suggesting a number of mixed effects in the results shown in Fig 5b.

Response: Thanks for this reviewer's comments. Although BSEP is not directly involved in bilirubin disposition, its malfunction or blockade disrupts normal bile flow thus, indirectly impacts bilirubin elimination through regulation of bile flow, resulting in elevated bilirubin content (10.1186/s12929-018-0475-8, 10.1021/mp4001348). We have provided explanation and comment on the use of BSEP inhibitor as mentioned by the reviewer and revised the corresponding text in the revised manuscript.

L225-227: 'MRP2 is a bottleneck in bilirubin excretion. In addition, it has been reported that malfunction or blockade of BSEP disrupts normal bile flow and thus adequate bilirubin clearance.'

- Finally, additional patient information is required. What cholestatic diseases were considered? A table with patient information, including the pathological diagnosis, is required. Is the pathology the same in all patients with CLI or within each severity group? Moreover, a high bilirubin level is not always a valid indicator of cholestasis, as it is also elevated in severe non-cholestatic liver diseases. Considering both bilirubin and ALP or GGT levels is likely a better approach for patient stratification. It is also important to note that table s4 has not been mentioned in the text.

Response: Thank you for the reviewer's comments. The Table S3 containing patient information (including pathological diagnosis) has been reorganized. The pathological diagnosis was made by gastroenterologists of the Affiliated Hospital of Zunyi Medical University based on the patient's clinical manifestations, signs, biochemical indicators, imaging examination and gastroenteroscopy. Samples were collected mainly from patients with cirrhosis and liver insufficiency combined with cholestasis, but also included patients with other diseases combined with cholestasis (such as gastrointestinal bleeding, gastrointestinal diseases, etc.). The criteria of inclusion of

samples were based on the clinical diagnosis of cholestasis. We agree that a high bilirubin level is not always a valid indicator of cholestasis. A combination of multiple factors such as bilirubin, ALP, and GGT may be a better indication for patient stratification. However, given the limited sample size, in this revised manuscript, we reclassified the clinical samples into normal group and patients instead of stratifying patients into mild and severe patients. This processing approach does not affect our judgment of the overall results. Stratification is just to better represent this phenomenon. We highly value the reviewer's opinions, and will adopt more detailed stratified designs that consider multiple factors in subsequent studies. In addition, Table S4 is mentioned in line 263 and highlighted in red in the revised manuscript.

Reviewer #2 (Remarks to the Author):

In the present investigation, Wang et al. investigated the role of Nrf2/HO-1 in cholestatic liver injury (CLI). They found that Nrf2/HO-1 activation enhances liver injury rather than protects from CLI with excessive bilirubin accumulation. Inhibiting HO-1 or ameliorating bilirubin transport alleviates liver injury in CLI models. In addition, Nrf2 knockout resulted in hepatoprotection in CLI mice. In contrast, Nrf2 knockout aggravated liver damage in non-CLI mice. In CLI, Nrf2/HO-1 induces bilirubin accumulation and impairs mitochondrial function. Furthermore, the expression of Nrf2 and HO-1 was elevated in the serum of patients with CLI, and the HO-1 expression was correlated with the severity of CLI and bilirubin levels.

The study by Wang et al. is interesting, well-written, and provides intriguing insights into the role of the Nrf2/HO-1 Nrf2/HO-1 axis in the liver. The experiments were properly and logically conducted. The data are solid and support the conclusions drawn.

Minor issue:

- The histology pictures are minuscule and hard to evaluate. New, more detailed, and bigger pictures should be provided at higher magnification.

Response: We appreciate the valuable feedback and questions raised by this reviewer. To better present the pathological changes of the liver, in this revised manuscript, we have provided larger or higher magnification, higher resolution histopathological images to make our results convincing. In addition, different liver pathological injury models are marked in the histological images. Please refer to Figure 1a, 2d, 3b, 4b, 4f,

S1b, and S5a. And all the changes are highlighted in red in the legends. Below are the figures (1a, 2d, 3b, 4b, 4f, S1b, S5a) and related legends.

Fig. 1a Representative H&E staining of liver tissues in mice (6- to 8-week-old) after administration of ZnPP, followed by 295.9 mg/kg OA or 60 mg/kg ANIT (n = 5). Neutrophil infiltration (green arrows) and hepatocyte necrosis (yellow arrows or dashed circles). Scale bar, 100 μ m or 50 μ m.

Fig. 2d Representative H&E staining of liver tissues. Neutrophil infiltration (green arrows) and hepatocyte necrosis (yellow arrows). Scale bar, 100 μ m (n = 5-6).

Fig. 3b

Fig. 3b Representative H&E staining of liver tissues in mice after administration of 1 mg/kg Fer-1 and 295.9 mg/kg OA. **Neutrophil infiltration (green arrows)** and **hepatocyte necrosis (yellow arrows)**. Scale bar, 50 μm (n = 6).

Fig. 3e

Fig. 3e Representative H&E staining of liver tissues in mice after administration of 20 mg/kg UDPGA and 295.9 mg/kg OA. **Neutrophil infiltration (green arrows)** and **hepatocyte necrosis (yellow arrows)**. Scale bar, 50 μm (n=5-6).

Fig. 4b

Fig. 4b Representative H&E staining of liver tissues (scale bar, 100 μm). **Neutrophil infiltration (green arrows).**

Fig. 4f

f Representative H&E staining of liver tissues (scale bar, 100 μm). **Neutrophil infiltration (green arrows) and hepatocyte necrosis (yellow arrows).**

Fig. S1a

Fig. S1a Representative images of H&E staining (scale bar, 100 μ m). **Ballooning degeneration (black arrows) and hepatocyte necrosis (yellow dashed circles).**

Fig. S5a

Fig. S5a Representative images of liver gross examination and H&E staining (scale bar, 100 μ m). **Ballooning degeneration (black arrows).**

We added the following sentences in Results:

Figure 1a (L105-106, L130-137): ‘OA induced extensive hepatocyte necrosis, nucleolysis and neutrophil infiltration.’ and ‘Multiple focal hepatocyte necrosis and neutrophil infiltration were observed in the ANIT group, ……and the hepatotoxicity induced by ANIT was significantly alleviated after administration of ZnPP, manifested by an improved morphology showing well-arranged hepatic lobules and hepatic cords with only a small amount of inflammation and hepatocyte necrosis, …….’

Figure S1a (L143-144, L149-150): ‘CCl₄ caused degeneration and necrosis of cells in the portal area or around the central vein.’ and ‘increased hepatocellular necrotic areas and extensive ballooning degeneration.’

Figure S5 (L178): ‘extensive ballooning degeneration’.

Reviewer #3 (Remarks to the Author):

COMMSBIO-23-3439-T

Activation of Nrf2/HO-1 signaling pathway exacerbates cholestatic liver injury

This is a research article that tries to explain the toxic functions of Nrf2/HO-1 signaling pathway in the hepatocyte cholestatic injuries. The manuscript consists of many experiments separately performed in both mice and cultured cells. It sounds energetic but includes some fundamental issues that undermine reliability of the study. Further, the text is too redundant to go through and understand.

1. The results of mice experiments, consisting most parts of the study and generating an initial question, are not very convincing because all the photomicrographs are too low qualities to evaluate the toxic or protective effects of OA and others. No pathological criteria of hepatocyte edema and large-scale necrosis is defined clearly. What kind of inflammatory cells are infiltrated in the liver? In such a study of experimental toxicology, histological evaluation would be very essential, and therefore, should be very clearly presented as first materials for the progression of experiments.

Response: We appreciate the valuable comments and questions raised by this reviewer. We have thoroughly edited our manuscript and made extensive changes to make the text flow more smoothly. Regarding the pathological changes, in the revised manuscript, we have replaced the images with larger or higher magnification, higher resolution images. Meanwhile, different liver pathological injury models are labeled and explained, as shown in Figure 1a, 2d, 3b, 4b, 4f, S1b and S5a. Explanations and changes are mentioned in legends and highlighted in red in the revised manuscript. The related figures (1a, 2d, 3b, 3e, 4b, 4f, S1b, S5a) and legends are mentioned above (Please refer to the response to Reviewer # 2).

We added the following sentences in Results in the revised manuscript:

Figure 1a (L105-106, L130-137): ‘OA induced extensive hepatocyte necrosis, nucleolysis and neutrophil infiltration.’ and ‘Multiple focal hepatocyte necrosis and neutrophil infiltration were observed in the ANIT group, ……and the hepatotoxicity induced by ANIT was significantly alleviated after administration of ZnPP, manifested by an improved morphology showing well-arranged hepatic lobules and hepatic cords with only a small amount of inflammation and hepatocyte necrosis, …….’

Figure S1a (L143-144, L149-150): ‘CCl₄ caused degeneration and necrosis of cells in the portal area or around the central vein.’ and ‘increased hepatocellular necrotic areas and extensive ballooning degeneration.’

Figure S5 (L178): ‘extensive ballooning degeneration’.

2. What is the meaning to measure the expression of Nrf2, a transcriptional factor, in the patient serum? Where does the Nrf2 expression come from?

Response: Thanks for this reviewer’s comments. Nrf2 can directly regulate the expression of HO-1 through binding to its promoter. In fact, we expected to detect the RNA expression of Nrf2 and HO-1 in human liver. However, due to the limitation and availability in sample collection, we hope to reflect the changes in human Nrf2 and HO-1 by detecting the RNA expression of Nrf2 and HO-1 in whole blood rather than liver tissue. The Nrf2 expression comes from white blood cells in the blood.

3. The results section is too long and very annoying to understand the experimental strategy and significance of the results. The results should be concisely described.

Response: We appreciate the valuable feedback and questions raised by this reviewer. We have reorganized the manuscript and made revisions to the results section of this article. These changes will not influence the content and framework of the paper. Here, we have not listed the changes but marked them in red in the revised manuscript.

4. Most of the results obtained are phenomenological but mechanistic.

Response: Thanks for this reviewer’s comments. In this study, we report the novel finding that activation of Nrf2/HO-1 exacerbates cholestatic liver injury rather than providing hepatic protection. We studied the effects of Nrf2/HO-1 signaling activation on the liver in two different animal models. In order to further verify the research results, we also used a CCl₄-challenged non-cholestatic liver injury model to detect histopathological changes and related biochemical indicators, and compared them with the cholestatic liver injury model. We believe these characterizations are necessary because this is the first report.

Nonetheless, we did conduct a series of experiments to understand the molecular

mechanisms underlying the effects of Nrf2/HO-1 activation on cholestatic liver injury. Mechanistic studies were performed in in vivo and in vitro experiments. In the in vivo experiments, we utilized Nrf2 knockout model to characterize the relevant mechanisms. In addition, different interventions (activators and inhibitors) were used to dissect the relevant mode of actions. In the in vitro experiments, we explored the molecular mechanisms of bilirubin-mediated mitochondrial dysfunction using genetic and pharmacological manipulations. We agree with this reviewer that we need to further explore the mechanisms and, in fact, we have been conducting more mechanistic studies related to the present project, which we hope to publish in the near future.

5. Some parts of the discussion are merely repeated description of the results. It does not include a discussion or show only too speculative comments.

Response: We appreciate the valuable comments raised by this reviewer. We have revised the whole content of this manuscript including the Discussion section. Particularly, we have deleted and modified some result descriptions in the Discussion section to avoid duplication in the revised manuscript. In addition, we have edited and added some discussions: (1) L333-336, ‘We demonstrate that bilirubin accumulation is ’; (2) L373-378, ‘Despite the impact of bilirubin on mitochondrial function, ’; (3) L381-388, ‘It is worth noting that emerging evidence suggests that HO-1 may induce ferroptosis through iron accumulation or other unknown mechanisms ’; (4) L395-401, ‘This seems to contradict the previous results showing that ’. Please refer to the discussion section for details.

We appreciate the editor and reviewers for editing and reviewing our manuscript.

Sincerely,

Shaoyu Zhou, Ph.D., Professor

Reviewers' comments:

Reviewer #1 (Remarks to the Author):

Authors satisfactorily answered most of my concerns.
However, only vague answers were given to some of my questions.

Among them, I consider particularly important to test the regulation of biliary transporters by Nrf2, as Nrf2 is key in this study (see title) and biliary transporters are key in cholestasis.

Authors agree with me that transporters also play an important role that may be entangled in the mechanism. They, indeed, measured/tested not only bilirubin but also bile acid transporters in several parts of the study. Authors demonstrated that the major bile acid and bilirubin transporters were downregulated in the OA model both in vivo and in vitro (Fig. 4a, Fig. S3). Moreover, they tested the role of the bile acid transporter BSEP by using a BSEP inhibitor in Figure 5.

Testing the effect of Nrf2 in biliary transporters could certainly strengthen the conclusions of this study. I encourage authors to test this possibility in the model / samples of their choice. I do not expect they to do new time-consuming experiment but to test the expression levels of transporters, by RT-PCR or western blot, in already existing samples where Nrf2 is KO or downregulated.

Reviewer #2 (Remarks to the Author):

The authors have convincingly addressed the concerns raised by the reviewer.

Reviewer #3 (Remarks to the Author):

The reviewer's questions were not answered clearly by the authors. Most parts of the study are still very phenomenological and merely showing results of small pieces that are difficult to be together for the discussion of pathogenesis. Moreover, some experimental strategy does not sound scientific.

1. Some of the figures become better resolution but not enough and the histopathological criteria is not described in the Material and Methods section.
2. What does it mean to measure the Nrf2 expression in the white blood cells for the study of liver injury? The experimental strategy is quite incomprehensive.
3. The sections of Results and Discussion is too redundant.

Reviewers' comments:

Reviewer #1 (Remarks to the Author):

Authors satisfactorily answered most of my concerns.

However, only vague answers were given to some of my questions.

Among them, I consider particularly important to test the regulation of biliary transporters by Nrf2, as Nrf2 is key in this study (see title) and biliary transporters are key in cholestasis.

Authors agree with me that transporters also play an important role that may be entangled in the mechanism. They, indeed, measured/tested not only bilirubin but also bile acid transporters in several parts of the study. Authors demonstrated that the major bile acid and bilirubin transporters were downregulated in the OA model both in vivo and in vitro (Fig. 4a, Fig. S3). Moreover, they tested the role of the bile acid transporter BSEP by using a BSEP inhibitor in Figure 5.

Testing the effect of Nrf2 in biliary transporters could certainly strengthen the conclusions of this study. I encourage authors to test this possibility in the model / samples of their choice. I do not expect they to do new time-consuming experiment but to test the expression levels of transporters, by RT-PCR or western blot, in already existing samples where Nrf2 is KO or downregulated.

Reviewer #2 (Remarks to the Author):

The authors have convincingly addressed the concerns raised by the reviewer.

Reviewer #3 (Remarks to the Author):

The reviewer's questions were not answered clearly by the authors. Most parts of the study are still very phenomenological and merely showing results of small pieces that are difficult to be together for the discussion of pathogenesis. Moreover, some experimental strategy does not sound scientific.

1. Some of the figures become better resolution but not enough and the

histopathological criteria is not described in the Material and Methods section.

2. What does it mean to measure the Nrf2 expression in the white blood cells for the study of liver injury? The experimental strategy is quite incomprehensive.

3. The sections of Results and Discussion is too redundant.

We would like to express our sincere appreciation to the editor and the reviewers for the valuable and constructive comments concerning our article entitled "activation of Nrf2/HO-1 signaling pathway exacerbates cholestatic liver injury". According to the editor and reviewers' comments, we have made further modifications to our manuscript and supplemented extra data (please refer to one-by-one responses below). In this revised version, all changes are highlighted within the document by using red-colored text.

The following are our one-by-one responses to the reviewers' comments.

Reviewer #1 (Remarks to the Author):

Authors satisfactorily answered most of my concerns.

However, only vague answers were given to some of my questions.

Among them, I consider particularly important to test the regulation of biliary transporters by Nrf2, as Nrf2 is key in this study (see title) and biliary transporters are key in cholestasis.

Authors agree with me that transporters also play an important role that may be entangled in the mechanism. They, indeed, measured/tested not only bilirubin but also bile acid transporters in several parts of the study. Authors demonstrated that the major bile acid and bilirubin transporters were downregulated in the OA model both in vivo and in vitro (Fig. 4a, Fig. S3). Moreover, they tested the role of the bile acid transporter BSEP by using a BSEP inhibitor in Figure 5.

Testing the effect of Nrf2 in biliary transporters could certainly strengthen the conclusions of this study. I encourage authors to test this possibility in the model / samples of their choice. I do not expect they to do new time-consuming experiment but to test the expression levels of transporters, by RT-PCR or western blot, in already existing samples where Nrf2 is KO or downregulated.

Response: We appreciate the valuable comments and suggestions by the reviewer. We agree with the reviewer that testing the effect of Nrf2 on transporter expression levels enhances mechanistic understanding of the role of Nrf2/HO-1 signaling activation in CLI. Therefore, we determined the mRNA expression levels of Bsep and Mrp2 in the

liver of wildtype and Nrf2 knockout mice challenged with OA. The new supplementary data have been added to the revised manuscript as Fig. S7 (shown below). The description of the new results is added to lines 214-217, and reads as follows: “We measured the expression of efflux transporters in the liver of WT mice and *Nrf2*^{-/-} mice to further investigate the role of efflux transporters. The results showed that OA could reduce the mRNA expression of efflux transporters (Bsep and Mrp2) in WT mice, but there were no significant changes in *Nrf2*^{-/-} mice (Fig. S7).”

In addition, we extended our discussion on the mechanism of Nrf2-mediated regulation of transports in relation to the newly added results. The newly added discussion is located in lines 338-348, and reads as follows: “However, the dysregulation of exporter transporters (BSEP and MRP2) may also affect the accumulation of bile acids, one of the important players in the pathology of CLI. Indeed, we found that Bsep and Mrp2 were downregulated in the liver of CLI mice in a Nrf2 dependent manner (Fig. S7). Although the mechanism of Nrf2-mediated downregulation of Bsep and Mrp2 has yet to be further investigated, these results suggest a role of bile acid regulation in the pathogenesis of liver injury, which is entangled in the exacerbation of liver injury induced by Nrf2/HO-1 activation in CLI. In this regard, further dissecting the molecular mechanism of Nrf2-mediated regulation of BSEP and MRP2 and its role in bilirubin accumulation may be necessary for comprehensively understanding the mechanism by which activation of Nrf2/HO-1 signaling pathway aggravates cholestatic liver injury.”

Figure S7. Nrf2/HO-1 mediated mRNA expression of *Bsep*, *Mrp2* in the liver of mice with OA-induced cholestatic liver injury (n = 5-6). Student's t test was used. The data are shown as the Mean \pm SEM. * $p < 0.05$, ** $p < 0.01$, significant difference compared to the control group., BSEP, bile salt export pump; MRP, multidrug resistance protein.

Reviewer #2 (Remarks to the Author):

The authors have convincingly addressed the concerns raised by the reviewer.

Response: Thank you for the reviewer's comments.

Reviewer #3 (Remarks to the Author):

The reviewer's questions were not answered clearly by the authors. Most parts of the study are still very phenomenological and merely showing results of small pieces that are difficult to be together for the discussion of pathogenesis. Moreover, some experimental strategy does not sound scientific.

Response: We appreciate the valuable feedback and questions raised by the reviewer. We agree with the reviewer that further mechanistic studies are of great significance. For example, in the context of cholestatic liver injury, what are the specific targets of bilirubin in liver mitochondria? Is there an interaction between bile acids and bilirubin that affects bilirubin accumulation and its hepatotoxicity? These and many other issues may be relevant to the results reported in this study, however, we believe that all merit further investigation in new projects.

1. Some of the figures become better resolution but not enough and the histopathological criteria is not described in the Material and Methods section.

Response: We thank the reviewer for the valuable comments and constructive suggestions. In this revised manuscript, we have added the following description in the Method section (L549-558): "Images were captured by using a light microscope (Olympus, Tokyo, Japan) to analyze liver damage. Descriptions of pathological changes of liver tissues included, but were not limited to, necrosis, inflammation, ballooning, etc. Typical lesion areas in the image are marked with arrows or dashed circles in the report. Under light microscopy, necrosis manifests as detectable hepatocyte pyknosis, nuclear fragmentation, and karyolysis, which is indicated by yellow arrows or dashed circles in the images. Neutrophil infiltration can be seen under light microscopy, and the cell nuclei are dark purple, which is indicated by green arrows or dashed circles in

the image. Under light microscopy, ballooning degeneration manifests as hepatocytes swelling into spherical shapes with almost transparent cytoplasm, which is indicated by black arrows in the image.”

Meanwhile, the image in Figure S5a has been replaced with a new image (shown below). The lesion areas of Figure S5a are marked and described (shown below). All changes are highlighted within the document by using red-colored text.

Fig. S5a

Figure S5. Expression of Nrf2/HO-1 mediated antioxidant proteins in liver mice with CCl₄-induced liver injury. a Representative images of liver gross examination and H&E staining (scale bar, 100 μ m). **Ballooning degeneration (black arrows).**

2. What does it mean to measure the Nrf2 expression in the white blood cells for the study of liver injury? The experimental strategy is quite incomprehensive.

Response: Thanks for the reviewer’s comments. Blood-based gene expression has been widely used to study liver injury, which may reflect the behavior of oxidative stress genes in liver tissue and disease status. For example, studies of liver tissue damage in patients with chronic hepatitis C have shown that a cluster of genes is expressed in a similar manner in the liver and blood using total RNA extracted from liver tissues and blood samples, respectively (doi.org/10.1007/s00705-020-04564-z). In our studies, we examined the expression of Nrf2 and HO-1 in the blood samples from patients with cholestasis diseases to correlate with other biochemical indicators. However, we did not examine the expression of Nrf2 and HO-1 in the liver due to unavailability of the liver tissues.

3. The sections of Results and Discussion is too redundant.

Response: We thank the reviewer for the valuable comments and suggestions. We have further revised the Result and Discussion section to avoid redundant statements and make the text flow more smoothly. All the changes are highlighted within the document by using red-colored text. Specific changes/modifications are as follows:

Results:

- (1) Section “**Increased HO-1 expression plays a role in pathological changes in CLI**”: We have deleted the last two sentences in this section (“We postulated that in the CCl₄-induced liver injury model, the bilirubin synthesized by HO-1 can be normally excreted and maintained at a level with antioxidant activity. When inhibiting the expression of HO-1, the antioxidant effect in the body is weakened, and ultimately the liver toxicity is enhanced”).
- (2) Section “**Inhibiting hepatobiliary transport leads to bilirubin accumulation and aggravates CLI**”: We have modified “Hepatobiliary transport dysfunction is considered as the major cause of cholestasis, and regulation of proteins such as bile salt export pump (BSEP) and multidrug resistance protein (MRP) plays essential role in the maintenance of bile secretion and enterohepatic circulation” at the beginning of the section to “Regulation of hepatobiliary transport proteins such as bile salt export pump (BSEP) and multidrug resistance protein (MRP) plays essential role in the maintenance of bile secretion and enterohepatic circulation”. And added new results “We measured the expression of efflux transporters in the liver of WT mice and Nrf2^{-/-} mice to further investigate the role of efflux transporters. The results showed that OA could reduce the mRNA expression of efflux transporters (Bsep and Mrp2) in WT mice, but there were no significant changes in Nrf2^{-/-} mice (Fig. S7)”.
- (3) Section “**Excessive accumulation of bilirubin causes mitochondrial impairment**”: We have deleted the last sentence “These results indicate that high level of bilirubin interfered with mitochondrial function and at the same time caused a large amount of ROS release, thus activating oxidative stress response, and ultimately leading to cellular injury”.

- (4) Section “**Mutual regulation between Nrf2/HO-1 activation and bilirubin accumulation**”: We have removed the outlines of individual experiments included in this section: “(1) to clarify the regulatory role of bilirubin on Nrf2/HO-1 signaling pathway; (2) to detect the effect of antioxidant resveratrol on bilirubin toxicity; and (3) to explore the effect of overexpression of mitochondrial-targeted Mn-SOD (SOD2) on bilirubin toxicity”.

Discussion:

- (1) Second paragraph: we modified “In the present study, we demonstrate that the accumulation of bilirubin induced by activation of Nrf2/HO-1 is hepatotoxic, which aggravates liver injury in CLI models. We not only found that the inhibition of HO-1 significantly alleviated liver injury, but also showed that Nrf2 knockout had a hepatoprotective effect in CLI. On the other hand, *Nrf2*^{-/-} mice exhibited increased liver damage in the CCl₄ induced liver injury model. Therefore, the activation of Nrf2/HO-1 is harmful in CLI but may have a protective effect in other non-cholestatic liver injury models” to “In the present study, we not only demonstrated that the Nrf2/HO-1 activation mediated bilirubin accumulation is hepatotoxic in CLI, but also showed that the inhibition of HO-1 or Nrf2 knockout conferred a hepatoprotective benefit in the setting of CLI. On the contrary, *Nrf2*^{-/-} mice exhibited increased liver damage in the CCl₄ induced non-CLI model”.
- (2) Third paragraph: We modified “We demonstrate that bilirubin accumulation is a critical player in exacerbation of cholestatic liver injury mediated by Nrf2 activation. In fact, cholestasis, especially chemical induced cholestasis, usually leads to impaired bilirubin excretion³⁹. One of the mechanisms involves the dysregulation of export transporters” to “Our results revealed that bilirubin accumulation is a critical player in exacerbation of cholestatic liver injury mediated by Nrf2 activation, which is related to the regulation of export transporters. In fact, cholestasis, especially chemical induced cholestasis, usually leads to impaired bilirubin excretion³⁹.”
- (3) Third paragraph: We added “However, the dysregulation of exporter transporters (BSEP and MRP2) may also affect the accumulation of bile acids, one of the important players in the pathology of CLI⁴². Indeed, we found that

Bsep and Mrp2 were downregulated in the liver of CLI mice in a Nrf2 dependent manner (Fig. S7). Although the mechanism of Nrf2-mediated downregulation of Bsep and Mrp2 has yet to be further investigated, these results suggest a role of bile acid regulation in the pathogenesis of liver injury, which is entangled in the exacerbation of liver injury induced by Nrf2/HO-1 activation in CLI. In this regard, further dissecting the molecular mechanism of Nrf2-mediated regulation of BSEP and MRP2 and its role in bilirubin accumulation may be necessary for comprehensively understanding the mechanism by which activation of Nrf2/HO-1 signaling pathway aggravates cholestatic liver injury”.

Sincerely,

Shaoyu Zhou, Ph.D., Professor

REVIEWERS' COMMENTS:

Reviewer #1 (Remarks to the Author):

Authors satisfactorily addressed my concerns. They determined the expression levels of Bsep and Mrp2 in the liver of wild type and Nrf2 KO mice challenged with OA. They properly discussed the results.

Reviewer #3 (Remarks to the Author):

The authors have convincingly addressed the concerns raised by the reviewer.